# Clustering of single-cell multi-omics data with a multimodal deep learning method

Xiang Lin[1,4], Tian Tian [2,4], Zhi Wei [1] ✉ & Hakon Hakonarson[2,3]

Single-cell multimodal sequencing technologies are developed to simultaneously profile different modalities of data in the same cell. It provides a unique opportunity to jointly analyze multimodal data at the single-cell level for the identification of distinct cell types. A correct clustering result is essential for the downstream complex biological functional studies. However, combining different data sources for clustering analysis of single-cell multimodal data remains a statistical and computational challenge. Here, we develop a novel multimodal deep learning method, scMDC, for single-cell multi-omics data clustering analysis. scMDC is an end-to-end deep model that explicitly characterizes different data sources and jointly learns latent features of deep embedding for clustering analysis. Extensive simulation and real-data experiments reveal that scMDC outperforms existing single-cell single-modal and multimodal clustering methods on different single-cell multimodal datasets. The linear scalability of running time makes scMDC a promising method for analyzing large multimodal datasets.

Single-cell RNA sequence (scRNA-seq) profiles a high-resolution picture inside an individual cell. Based on scRNA-seq technology, recently, many multimodal sequencing technologies have been developed to jointly profile multiple modalities of data in a single cell. For example, cellular Indexing of Transcriptomes and Epitopes by Sequencing (CITE-seq) and RNA expression and protein sequencing assay (REAP-seq) have been developed to profile mRNA expression and quantify surface protein simultaneously at the cellular level[1,2]. Specifically, CITE-Seq employs existing single-cell sequencing technologies, such as the 10X Genomics Chromium platform[3], and allows the counting of Antibody-Derived Tags (ADT) to quantify the cell-surface protein abundance. Each cell with ADT labels and DNA-barcoded microbeads will be encapsulated in a droplet for single-cell sequencing[4]. REAP-seq also combines DNA-barcoded antibodies with existing scRNA-seq approaches to measure the expression levels of genes and cell-surface proteins[2]. In addition to studying single-cell transcriptomes and surface proteins, recently, the development of single-cell approaches for the assay of the transposase accessible chromatin sequencing (scATAC-seq) provides us a chance to measure chromatin accessibility in a single cell[5]. Specifically, these technologies are designed to identify open chromatin regions in the genome by using the hyperactive Tn5 transposase, which simultaneously tags and fragments DNA sequences in open chromatin regions[6]. The scATAC-seq enables us to explore cell type-specific biological activities by investigating the chromatin-accessibility signatures, such as the transcription factors that control the gene expression of cells. More recently, some multi-omics single-cell technologies have been developed to jointly profile chromatin accessibility and gene expression within a single cell[7], such as SNARE-seq and 10X Single-Cell Multiome ATAC + Gene Expression (we denote it as SMAGE-seq)[8,9]. Overall, these multimodal sequencing technologies provide us with a more comprehensive and complicated profile of a single cell. Therefore, the computational tools for jointly integrating different data views for downstream analyses, such as clustering analysis, are desired for these new powerful experimental technologies.

It is noted that in the multimodal data, the biological information provided by different modalities is complementary[2,4], and each modality generally has its own strengths and weaknesses. Using CITE-seq as an example, its ADT modal focuses on surface proteins. ADT data have demonstrated a low dropout rate[4] and thus can reliably quantify cell

[1]Department of Computer Science, New Jersey Institute of Technology, Newark, NJ, USA. [2]Center of Applied Genomics, Children's Hospital of Philadelphia, Philadelphia, PA, USA. [3]Division of Human Genetics, Department of Pediatrics, Perelman School of Medicine, University of Pennsylvania, Philadelphia, PA, USA. [4]These authors contributed equally: Xiang Lin, Tian Tian. ✉e-mail: zhiwei@njit.edu

activities. For the five CITE-seq datasets analyzed in this study, we observed dropout rates of up to 12% in ADT data. In contrast, there were more than 80% or even 90% zero entries in its corresponding mRNA data. For most genes, protein is the final product to fulfill their functions and messenger RNA is an immediate product. Thus, ADT data seems ideal for characterizing cell functions and types. However, due to current technique limits, ADT can profile only up to a couple of hundreds of proteins. Because of this limit, investigators generally include well-known cell type markers in ADT modal first. Therefore, ADT data is good at identifying common cell types[4,10], such as CD4+ and CD8+ T cells, when their marker genes are profiled. However, because of its limited dimensions, ADT data may not detect rare or minor cell types well. In contrast, the full transcriptome of mRNA data can capture comprehensive cell types. Nevertheless, clustering cells based on scRNA-seq may be challenged by its large dropout rate and sparse signal with high dimensionality. Furthermore, the quantity of ADT and mRNA sources produced by the same gene may not be the same when considering the post-transcriptional and post-translational regulations[4,11]. In this case, ADT and mRNA data provide complementary information in cell type identification[10]. For SNARE-seq and SMAGE-seq, scATAC-seq data provides chromatin accessibility information which is also complementary to mRNA data[8]. Thus, by integrating the information from multimodalities, we should be able to arrive at a higher resolution of cell typing.

Clustering analysis is an essential step in most single-cell studies and has been studied extensively. Based on the clustering results, researchers can explore the biological activities in cell type or subtype level, which could not be reached by studying bulk data[12–14]. Numerous clustering methods have been designed for the analysis of scRNA-seq data. For example, Tscan applies principal component analysis (PCA) on the scRNA-seq data and then performs the Gaussian mixture model (GMM) clustering on the low-dimensional representation[15]. Seurat constructs a k-nearest neighbors (KNN) graph based on the Euclidean distance in PCA space. With the graph, it then employs the Louvain[16]/Leiden algorithm to iteratively group cells together by optimizing modularity[17]. The Louvain/Leiden algorithm has already become one of the most popular methods for scRNA-seq clustering. SC3 employs spectral clustering to obtain individual clustering results based on the distance matrices derived from the Euclidean, Pearson and Spearman metrics, respectively. It then computes a consensus matrix by summarizing the three individual clustering results. Finally, the consensus matrix is clustered using hierarchical clustering to produce final clustering results[18]. However, these traditional single-cell clustering methods are not ready to take the advantage of multi-omics data to improve clustering performance and are thus not applicable to multimodal data.

A couple of methods have emerged for the clustering analysis of CITE-seq data in the past years. Recently, we proposed a single cell deep constrained clustering framework – scDCC that can integrate ADT information into the clustering analysis of scRNA-seq data by manually defined constraints[19]. BREM-SC[10], a hierarchical Bayesian mixture model, applies two multinomial models to jointly characterize scRNA-seq and ADT data. It assumes that the proportions (relative expression levels of genes or proteins) in the multinomial models follow Dirichlet distributions, and cell-specific random effects are introduced to model the correlation between the two data sources. Although BREM-SC is one of the first proposed models for clustering analysis of CITE-seq data, it has several limitations. Firstly, it assumes that the data follow a certain specific distribution. Such parametric assumptions may not hold in all real applications. Secondly, BREM-SC does not characterize the dropout events, which is the major problem in the clustering of scRNA-seq data. Finally, BREM-SC has a scalability issue. The running time of BREM-SC becomes costly and slow when analyzing thousands of cells.

Meanwhile, CiteFuse, Seurat V4, and Specter can cluster CITE-seq data by using distance-based graphs. CiteFuse[20] calculates the cell-to-

cell similarity matrices of ADT and mRNA separately and then merges them by a similarity network fusion algorithm[21]. Clustering is performed on the merged similarity matrix by using graph-based clustering algorithms such as spectral[22] and Louvain algorithm[16]. However, similarity matrix-based clustering cannot explicitly consider the dropout events in scRNA-seq data. Hao et al. developed a weighted nearest-neighbor (WNN) procedure in Seurat V4 for multi-omics data clustering[23]. Briefly, the WNN procedure learns the weights of multi-modal data and generates a similarity graph of cells by a weighted combination of mRNA and protein views. Van et al.[24] proposed a landmark-based spectral clustering (LSC) method, Spector, for clustering single-cell data with linear-time scalability. LSC picks a small set of cells as the landmarks and calculates a Gaussian kernel-based similarity matrix between the rest of the cells and the landmarks, then the whole Laplacian matrix is built. Different omics require a different choice of the number of landmarks and the kernel bandwidth, and consensus clustering is used for ensembles across modalities. Compared to BREM-SC and CiteFuse, the WNN algorithm and Specter run much faster and require less memory. However, these two methods fail to take into consideration the dropout events in the count data too.

Another line of research, which is relevant, focuses on learning a joint embedding of different modalities. Such joint embedding is expected to improve various downstream analyses, including clustering. TotalVI is a deep variational autoencoder that can capture the same latent space of different data types[25]. With this design, TotalVI can learn a joint probabilistic representation of the paired ADT and mRNA measurements from CITE-seq data that accounts for the distinct information of each modality. Similarly, for SNARE-seq or SMAGE-seq data, Cobolt[26] and scMM[27] employ a Multimodal Variational Autoencoder to jointly model the multiple modalities and learn a joint embedding of the single-cell mRNA-seq and ATAC-seq data. However, these methods focusing on joint embedding are not designed and optimized for clustering, although we can, as a naïve solution, learn joint embeddings first, which is then followed by simple clustering using, for example, k-means. Such a divided strategy is suboptimal for clustering, as shown in our experiments later.

As we mentioned above, many existing methods fail to consider the dropout events in the single-cell data during the learning of embedding and/or clustering. However, the pervasive dropout events make single-cell count data to be zero-inflated and over-dispersed. To better characterize single-cell mRNA count data, a zero-inflated negative binomial (ZINB) model has been widely used to account for the large dispersion and the dropout events[28,29]. Many ZINB model-based methods, including deep learning approaches, have been developed to analyze scRNA-seq count data, including ZINB-WaVE[29], DCA[30], scVI[31], and scDeepCluster[28], to name a few. These studies show that the ZINB model can effectively characterize scRNA-seq data and improve the representation learning and clustering results.

In this article, we propose a multimodal deep learning model, Single Cell Multimodal Deep Clustering (scMDC), for the clustering analysis of multimodal single-cell data. The network architecture of scMDC is shown in Fig. 1. scMDC employs a multimodal autoencoder[32], which applies one encoder for the concatenated data from different modalities and two decoders to separately decode the data from each modal. Following scDeepCluster[28], we apply ZINB loss as the reconstruction loss. The bottleneck layer is used for a deep K-means clustering[33]. To further improve latent feature learning, we introduce a Kullback-Leibler divergence-based loss (KL loss), which attracts similar cells and separates dissimilar cells[34]. The whole model, including the autoencoder, the KL-loss, and the deep K-means clustering, are optimized simultaneously. scMDC is an end-to-end multimodal deep learning clustering method for modeling different multi-omics data. Taking the advantage of graphics processing units (GPU), scMDC is very efficient in the analyses of large datasets. In addition, by employing a conditional autoencoder framework, scMDC can correct

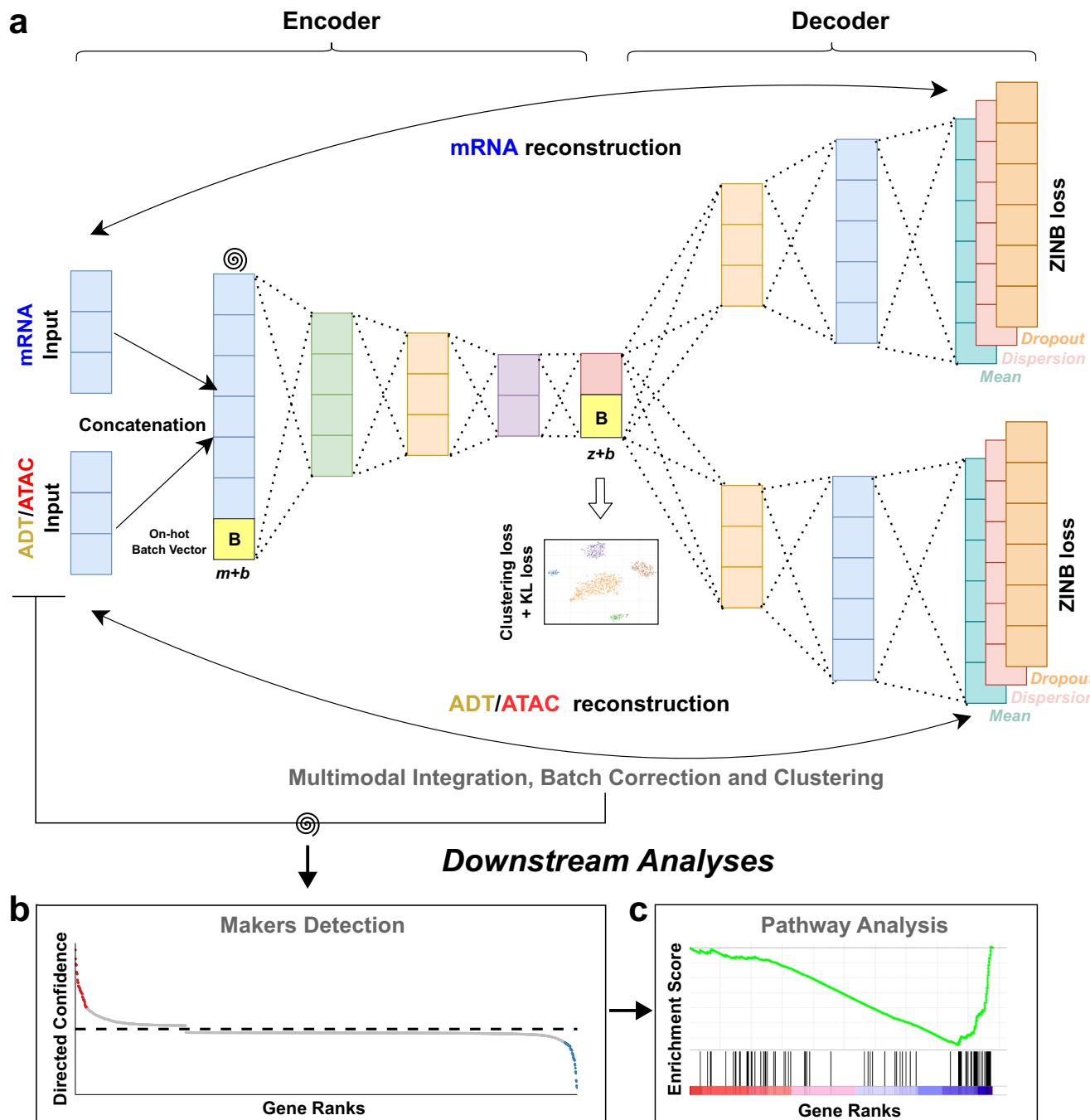

**Fig. 1 | The architecture of scMDC.** scMDC has one encoder for the concatenated data and two decoders for each modal in the multimodal data (**a**). It can be used for clustering CITE-seq data and 10x Single-Cell Multiome ATAC + Gene Expression (SMAGE-seq) data. The spiral symbols indicate the artificial noises added to the data. For multi-batch datasets, scMDC will work in a conditional autoencoder manner. A one-hot batch vector **B** (in dimension $b$) will be concatenated to the input feature of the encoder (with raw feature dimension, $m$) and the decoders (with latent feature dimension, $z$). This is designed for batch effect correction. scMDC learns a latent representation **Z** (in dimension $z$) of data on which different modalities are integrated. A deep K-means algorithm and a KLD loss are implemented on **Z**. Based on the clustering results, scMDC employs an ACE model[36] to detect markers in different clusters (**b**). Then, pathway analyses can be conducted based on the gene ranks learned by ACE (**c**).

batch effects when analyzing multi-batch data. To our knowledge, scMDC is the first end-to-end deep clustering method that can both integrate multimodal data and remove the batch effect for different types of multimodal data. The superior performance of scMDC is observed from the extensive experiments on both CITE-seq and SMAGE-seq data. After clustering, for a given cluster, we also detect the markers (genes or proteins) by transplanting an ACE model[35] to scMDC and conduct a gene set enrichment analysis based on the gene ranks learned from ACE. The meaningful results of these downstream

analyses further support the superior clustering performance of scMDC. We conclude that scMDC is a promising tool for clustering multimodal single-cell data.

## Results

### Real CITE-seq data evaluation

We first evaluate the clustering performance of scMDC on CITE-seq datasets in comparison with ten competing methods. The competing methods include the models designed for multimodal data clustering

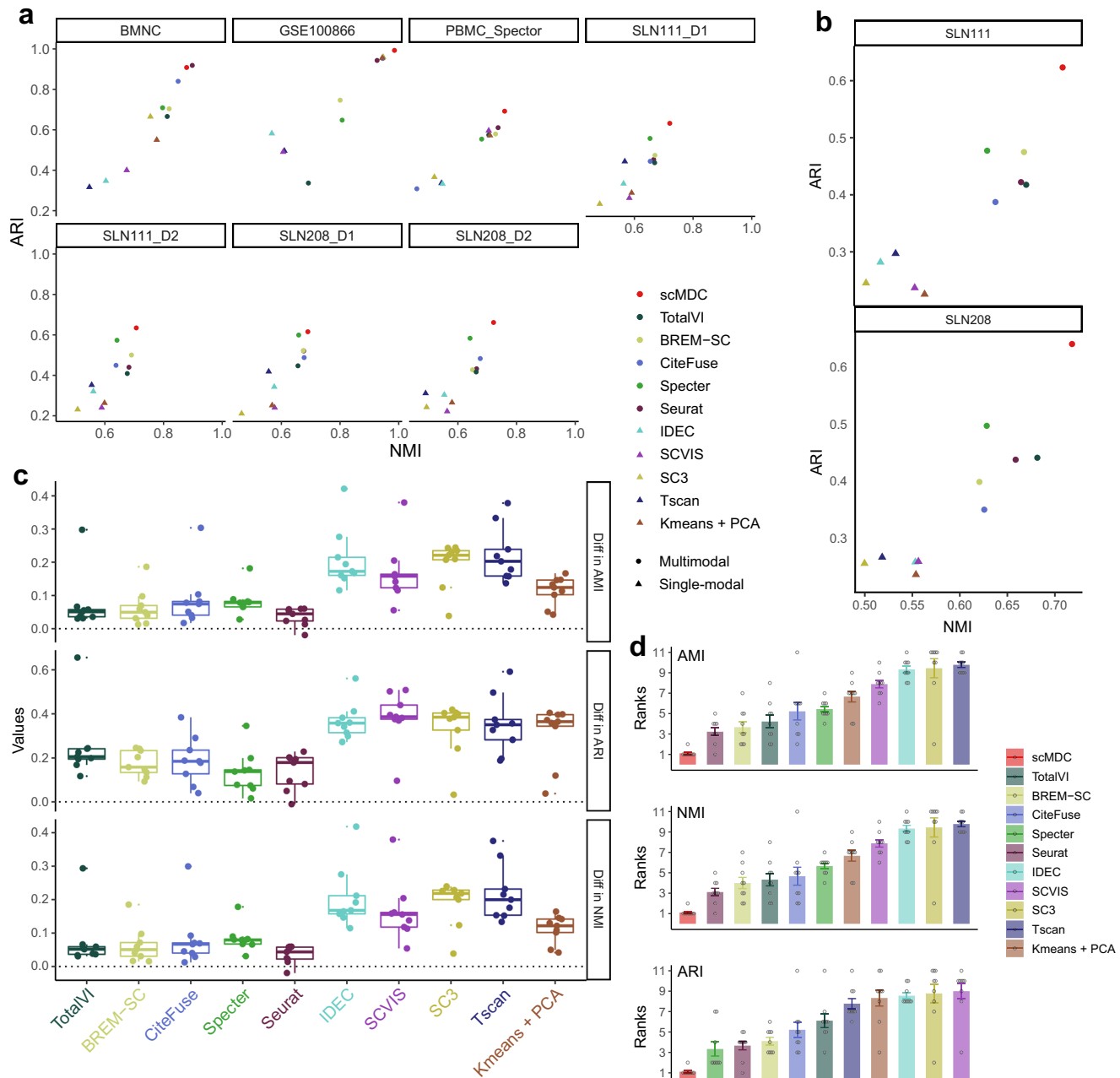

**Fig. 2 | Clustering performance of scMDC and the competing methods on different CITE-seq datasets.** All the methods are tested on seven one-batch datasets (**a**, *n* = 7) and two two-batch datasets (**b**, *n* = 2). In panels (**a**) and (**b**), clustering performance is illustrated in a two-dimensional manner with ARI as the Y axis and NMI as the X axis. Circles stand for the results of the multi-omics methods and triangles stand for the results of the single-omics methods. The differences between the performance of scMDC and the competing methods are shown in boxplots (**c**, *n* = 9). Each boxplot shows the minimum, first quartile (Q1), median,

third quartile (Q3), and maximum of data. The minimum and maximum are the smallest data point that is equal to or greater than Q1 −1.5*IQR and the largest data point that is equal to or less than Q3 + 1.5*IQR, respectively. Each data point (a difference of performance in a dataset) is shown by a dot. We also summarized the performance of each method by showing the averaged ranks (**d**, *n* = 9). Each data point (a rank of a method in a dataset) is shown by a dot and the standard errors are shown by the error bars. In panels (**c**) and (**d**), clustering performance is evaluated by AMI, NMI, and ARI. Source data is provided as a Source Data file.

(BREM-SC, CiteFuse, Specter, and SeuratV4), the models developed for learning an embedding for single or multimodal data (SCVIS and TotalVI), two clustering tools for single-cell data (SC3 and Tscan), and two general clustering methods (IDEC and K-means). We test these tools on seven single-batch CITE-seq datasets and two multi-batch CITE-seq datasets. Of these ten methods under comparison, only scMDC, Seurat, and TotalVI can correct batch effects before clustering. We hypothesize that scMDC can boost the clustering performance in all the CITE-seq real datasets. Figure 2 shows the performance (AMI,

NMI, and ARI) of all the methods for different datasets. Overall, the multimodal methods have shown clear advantages over the single-modal methods. As shown in Fig. 2a, scMDC has demonstrated superior performance over competing methods across two metrics for most single-batch datasets except the BMNC dataset, in which Seurat has comparable performance. For the two multi-batch datasets, scMDC outperforms all the competing methods (Fig. 2b); TotalVI and Seurat are inferior to scMDC but outperform the other competing methods, thanks to their capability of correcting batch effects. The

differences between the performance of scMDC and the competing methods are summarized in Fig. 2c. A positive difference means higher performance in scMDC than the competing methods. We find that scMDC has a steady advantage over all the competing methods in multiple datasets. We then rank all competing methods for each dataset based on their performance metrics. Figure 2d shows the averaged rank of each method for the nine datasets. We can see that scMDC constantly ranks number 1 in all datasets for all three metrics. In contrast, the second-best methods, Seurat for AMI and NMI and Specter for ARI, have an averaged rank of 3. Using one-sided paired *t*-tests on the clustering metrics (AMI, NMI, and ARI), we confirm that the improvements of scMDC over competing methods are all significant (Supplementary Table 1). In summary, our results on multiple real datasets reveal that scMDC has stable and robust clustering performance on the CITE-seq datasets.

## Real SMAGE-seq data evaluation

We then test the clustering performance of scMDC on the SMAGE-seq data. Here we compare scMDC with four competing methods: Cobolt, scMM, SeuratV4, and K-means + PCA. Cobolt and scMM are designed for multi-omics data embedding learning. SeuratV4 is developed for CITE-seq data but here we apply the WNN algorithm to the SMAGE-seq data. We test these methods on three real SMAGE-seq datasets from 10X genomics, including two PBMC datasets and one embryonic mouse brain dataset. We also conduct a multi-batch experiment by combining two PMBC datasets (denoted as PBMC13K). For scATAC-seq data, we use a cell-to-gene matrix as input for scMDC, scMM, Seurat, and Kmeans. This matrix is built by mapping ATAC reads onto the gene regions (See method for details). Cobolt uses the peak count matrix as the input. Figure 3 shows the clustering performance of scMDC and the competing methods in single-batch datasets (a) and multi-batch datasets (b). We find that scMDC has superior performance in both single- and multi-batch datasets from all the metrics (NMI and ARI). Cobolt is the second-best method in the tests and has a comparable performance with scMDC on the E18 dataset in NMI, but its performance is inferior to that of scMDC in other datasets. Figure 3c summarizes the differences in clustering performance between scMDC and the competing methods. We find that the median differences are around 0.1 in AMI and NMI, and around 0.3 in ARI for all the competing methods, which illustrates the superiority of scMDC. We then rank all competing methods for each dataset based on their performance metrics. Figure 3d shows the averaged rank of each method for the four datasets. We can see that scMDC ranks best in all three metrics, while Cobolt is the second-best for AMI and ARI, and Seurat is the second-best for ARI. Using one-sided paired *t*-tests done on the raw performance metrics, we confirm that the improvements of scMDC over competing methods are all significant (Supplementary Table 2).

Taking the results from CITE-seq and SMAGE-seq experiments together, we conclude that scMDC is a general and promising clustering model for various single-cell multimodal data.

## Simulation experiments

To test the robustness of scMDC under different scenarios, we conduct two simulation experiments with various clustering signals and dropout rates. We generate all the simulation datasets using the SymSim package (v0.0.0.9) in R. Figure 4a–c show the performance of scMDC and the competing methods on the simulated CITE-seq data with low, medium, and high clustering signals, respectively. scMDC has demonstrated superior performance across all levels of clustering signals, especially in terms of AMI and NMI. TotalVI has comparable performance with scMDC in ARI, but it is outperformed by scMDC in other metrics. Besides, when the clustering signal is low, scMDC shows a greater advantage over other methods, revealing its capability to handle datasets with low signal-to-noise ratios. Figure 4d–f show the clustering results of all the methods with low, medium, and high

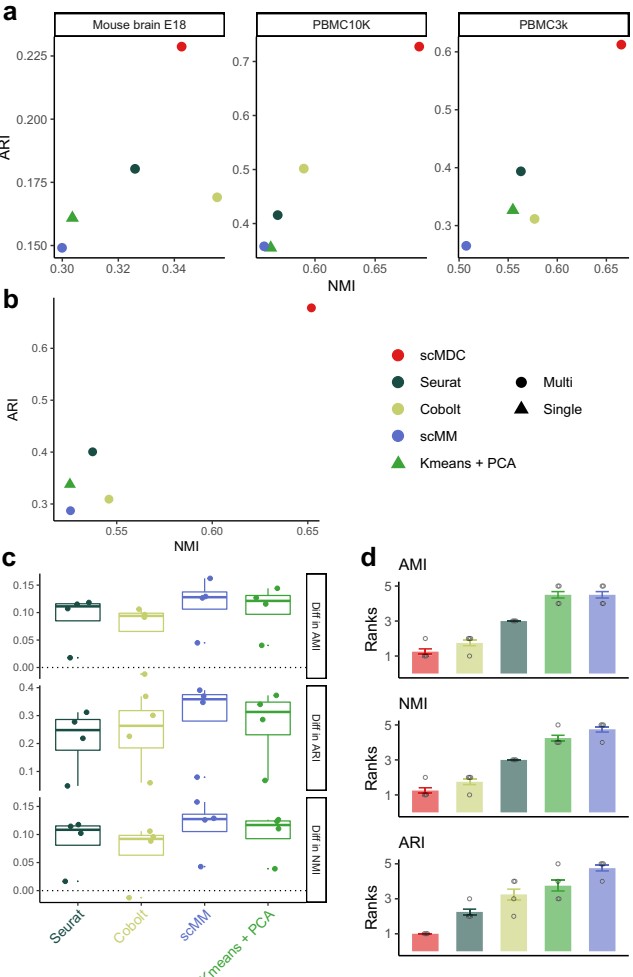

**Fig. 3 | Clustering performance of scMDC and the competing methods on different SMAGE-seq datasets.** All the methods are tested on three one-batch datasets (**a**, *n* = 3) and one two-batch dataset (**b**, *n* = 1). In panels (**a**) and (**b**), clustering performance is illustrated in a two-dimensional manner with ARI as the *Y*-axis and NMI as the *X*-axis. Circles stand for the results of the multi-omics methods and triangles stand for the results of the single-omics methods. The differences between the performance of scMDC and the competing methods are shown in boxplots (**c**, *n* = 4). Each boxplot shows the minimum, first quartile (Q1), median, third quartile (Q3), and maximum of data. The minimum and maximum are the smallest data point that is equal to or greater than Q1 − 1.5 * IQR and the largest data point that is equal to or less than Q3 + 1.5 * IQR, respectively. Each data point (a difference in performance in a dataset) is shown by a dot. We also summarized the performance of each method by showing the averaged ranks (**d**, *n* = 4). Each data point (a rank of a method in a dataset) is shown by a dot and the standard errors are shown by the error bars. In panels (**c**) and (**d**), clustering performance is evaluated by AMI, NMI, and ARI. Source data is provided as a Source Data file.

dropout rates, respectively. We can see that scMDC yields the optimal performance under various dropout rates, followed by TotalVI. We also observe that, the higher the dropout rate, the larger the improvement scMDC brings, in comparison with its competing methods. Such a result is compelling because most real single-cell datasets exhibit high dropout rates. The robust performance under high dropout events makes scMDC to be a superior clustering method. This result also consolidates our statement that scMDC is a better tool to cluster the datasets with low signal-to-noise ratios than the competing methods. For multi-batch data, we compare scMDC with TotalVI and Seurat, the only two competing methods that can correct batch effects. Medium dropout rate and clustering signal are used for simulating the multi-batch dataset. scMDC outperforms the two

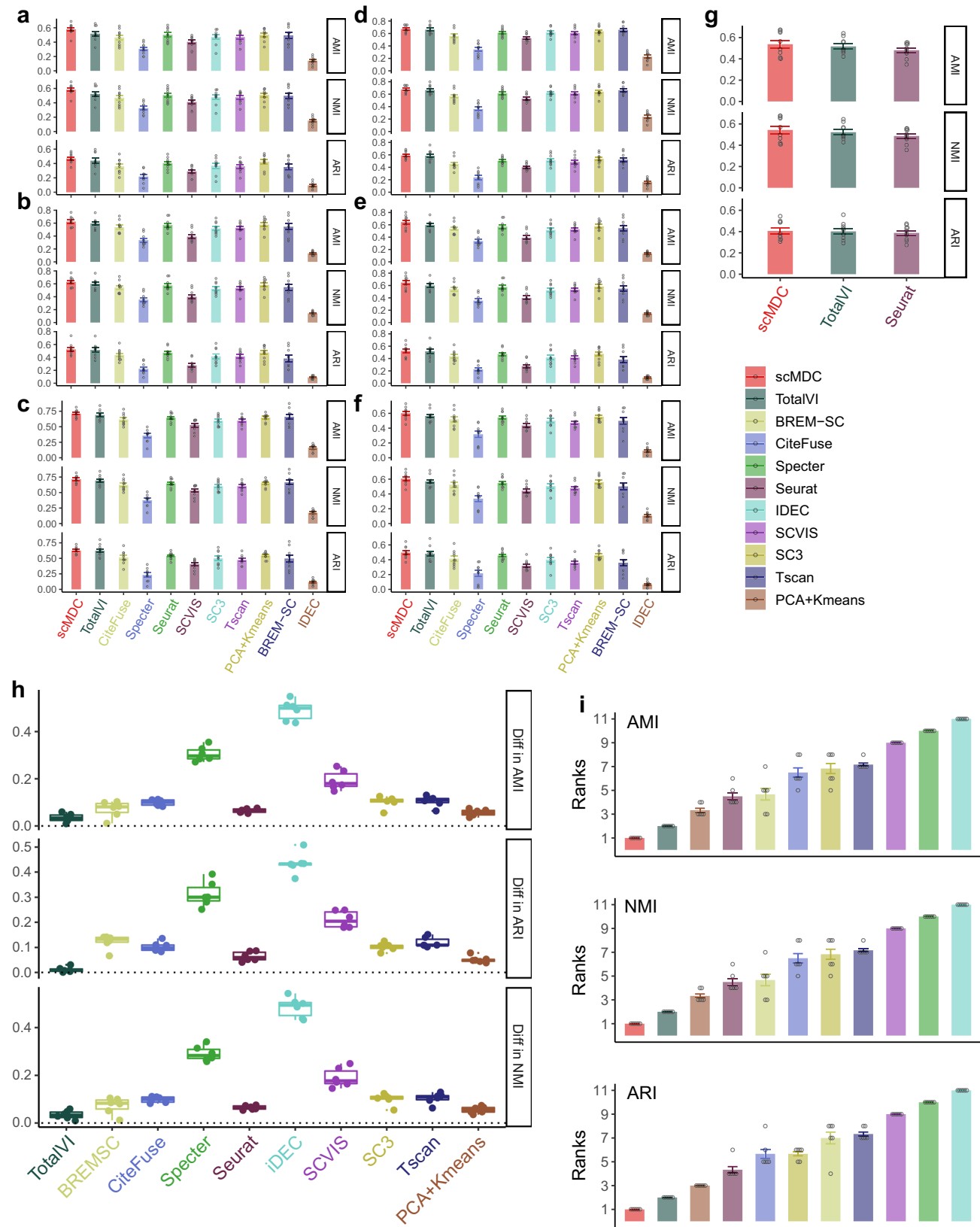

competing methods in all three metrics (Fig. 4g). The differences between the performance of scMDC and each competing method are summarized in Fig. 4h. Although the distribution of differences varies across different methods, all the medians of differences are greater than zero indicating a consistent superiority of scMDC over all the competing methods. Similarly, we rank all methods in the analyses of

these simulated datasets. scMDC and TotalVI constantly rank No. 1 and No. 2, respectively (Fig. 4i). Like the results in the real datasets, multi-omics methods have better overall performance than single-source methods. Using one-sided paired t-tests done on the three raw performance metrics, we confirm that the improvements of scMDC over competing methods are all significant (Supplementary Table 3). These

**Fig. 4 | Clustering performance of scMDC and the competing methods on the simulation datasets.** The first simulation experiment is to test the clustering performance of scMDC with low (**a**), medium (**b**), and high (**c**) clustering signals. The second simulation experiment is to test the clustering performance of scMDC with low (**d**), medium (**e**), and high (**f**) dropout rates. Since scMDC, Seurat, and TotalVI can correct the batch effect, we also test their clustering performance on a multi-batch simulation dataset (**g**). In panels (**a**–**f**), bars stand for the mean values, dots stand for the data points, and error bars stand for the standard errors. We generate ten replicates for each experimental setting ($n = 10$). The differences between the averaged performance of scMDC and the competing

methods over all simulation datasets are shown in boxplots (**h**, $n = 6$). Each boxplot shows the minimum, first quartile (Q1), median, third quartile (Q3), and maximum of data. The minimum and maximum are the smallest data point that is equal to or greater than Q1 − 1.5 * IQR and the largest data point that is equal to or less than Q3 + 1.5 * IQR, respectively. Each data point (a difference of averaged performance in a dataset) is shown by a dot. We also summarized the performance of each method by showing the averaged ranks (**i**, $n = 7$). Each data point (an average rank of a method in a setting) is shown by a dot and the standard errors are shown by the error bars. In all panels, the clustering performance is evaluated by AMI, NMI, and ARI. Source data is provided as a Source Data file.

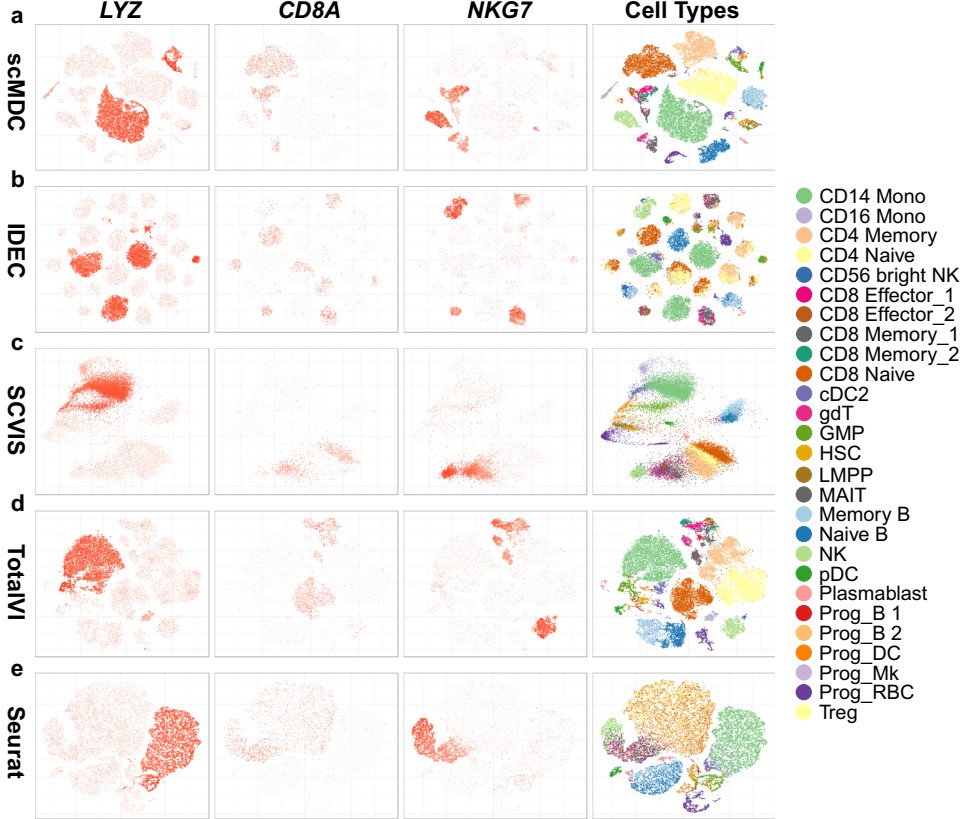

**Fig. 5 | Low-dimension representation of scMDC and the competing methods on the BMNC dataset.** The t-SNE plots of the embeddings from scMDC (**a**) and four competing methods including IDEC (**b**), SCVIS (c), TotalVI (**d**), and Seurat (**e**) are shown in different rows. The first three columns show the expression pattern of genes *LYZ*, *CD8A*, and *NKG7*. The last column shows the true labels (cell types) on the latent space learned from each method.

simulation results demonstrate that scMDC has robust clustering performance under various scenarios.

## Latent representations of real data

Figure 5 shows the t-SNE plots of the embedding of scMDC (a) and four competing methods, IDEC (b), SCVIS (c), TotalVI (d), and Seurat (e), on the BMNC dataset. We also show the expression pattern of three marker genes in the t-SNE plots. They are *LYZ* (the first column) for CD14 monocyte cells, *CD8A* (the second column) for CD8 cells, and *NKG7* (the third column) for NK cells. True labels (cell types) are shown in the fourth column. We find that scMDC can divide most cell types in the latent space. In contrast, SCVIS, TotalVI, and Seurat fail to separate many cell types, including some large cell types, such as CD14 monocyte and CD4 memory cells, which are connected or mixed with other cell types in the latent spaces. IDEC divides large cell types into many small clusters. Many of them are mixed with other cell types. It is noted that scMDC fails to divide some sub-cell types, such as CD8 effect 1, CD8 effect 2, CD8 memory 1, and CD8 memory 2, on the latent space. This problem is also observed on the t-SNE plots of other methods. In the latent space of scMDC, the marker genes are only expressed in

some isolated clusters. However, in the latent space of other methods, the marker genes are either expressed in multiple clusters or in a part of a large cluster. These are all unsatisfactory expression patterns. Similar results are observed in the expression pattern of ADT markers (Supplementary Fig. 1). We then build t-SNE plots of the embeddings of a multi-batch dataset SLN111 with two batches of data (Fig. 6). This dataset contains 28 cell types including some large ones (>1000 cells, such as CD4 and CD8 T cells) and tiny ones (<100 cells, such as erythrocytes and plasmacytoid dendritic cells). An ideal model should be capable of 1) dividing different cell types on the latent space, and 2) removing the batch effect and mixing the cells from different batches on the latent space. In other words, biological variations should be captured while technical variations are omitted during the embedding learning. Figure 6 shows the latent representations of scMDC (a) and four competing methods including IDEC (b), SCVIS (c), TotalVI (d), and Seurat (e). We find that scMDC can separate most cell types in the latent space. In addition, it mixes the cells from two batches in most clusters. IDEC can separate the large cell types but fails to divide many small cell types. SCVIS, TotalVI, and Seurat show inferior performance in dividing different cell types in the latent space. Like scMDC, TotalVI

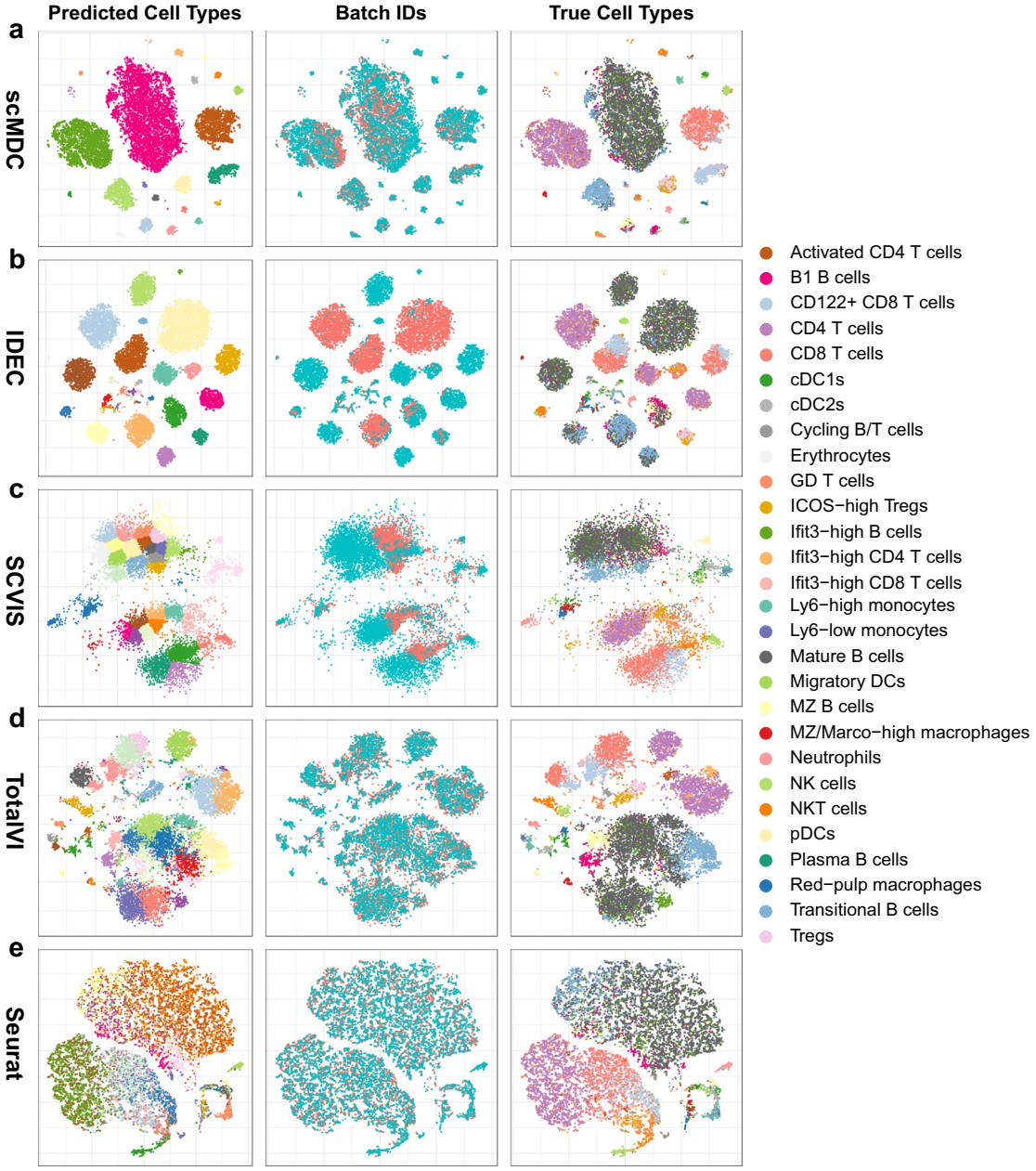

**Fig. 6 | Low-dimension representation of scMDC and the competing methods on the SLN111 dataset.** The t-SNE plots of the embeddings from scMDC (**a**) and four competing methods including IDEC (**b**), SCVIS (**c**), TotalVI (**d**), and Seurat (**e**) are shown in different rows. The three columns show the predicted labels, batch IDs, and true labels on the latent space learned from each method.

and Seurat also have satisfactory performance on batch effect correction. SCVIS and IDEC cannot address the batch effects, so the cells from the two batches are separated on the latent space. In summary, scMDC is the only method that has superior performance on both cell type partition and batch effect removal. Similar results can be found on the t-SNE plots of a multi-batch SMAGE-seq dataset (PBMC13K, Supplementary Fig. 2).

**The advantages of using multimodal data**
As described in the introduction, different omics of data provide different and complementary information for cell clustering and cell typing. Therefore, using multi-omics data in clustering should be able to achieve better performance than using single-source data. In this experiment, we conduct two tests. In the first test, we compare the performance of scMDC with three variant models: a sub-model of scMDC with only mRNA input and reconstruction loss (named scMDC-

RNA), a sub-model of scMDC with only ADT/ATAC input and reconstruction loss (named scMDC-ADT/scMDC-ATAC), and a variant model with concatenated mRNA and ADT data as input but with only one reconstruction loss (named as scMDC-Concat). Figure 7a, b shows the performance of scMDC and three variant models in CITE-seq and SMAGE-seq data, respectively. We find that scMDC outperforms the variant models in all the datasets. For CITE-seq data, scMDC-ADT has the second-best performance in all datasets. This is consistent with our expectation because most ADTs are strong markers for identifying some cell types. On the other hand, scMDC-ATAC has inferior performance in two SMAGE-seq datasets. The differences between the performance of scMDC and each variant model are summarized in Fig. 7c. We find a stable advantage of scMDC over all the variant models. Using a one-sided paired *t*-test, we find that scMDC significantly outperforms most variant models for both CITE-seq and SMAGE-seq data (Supplementary Table 4). The only exception is the scMDC-ATAC model (*P*-value = 0.07),

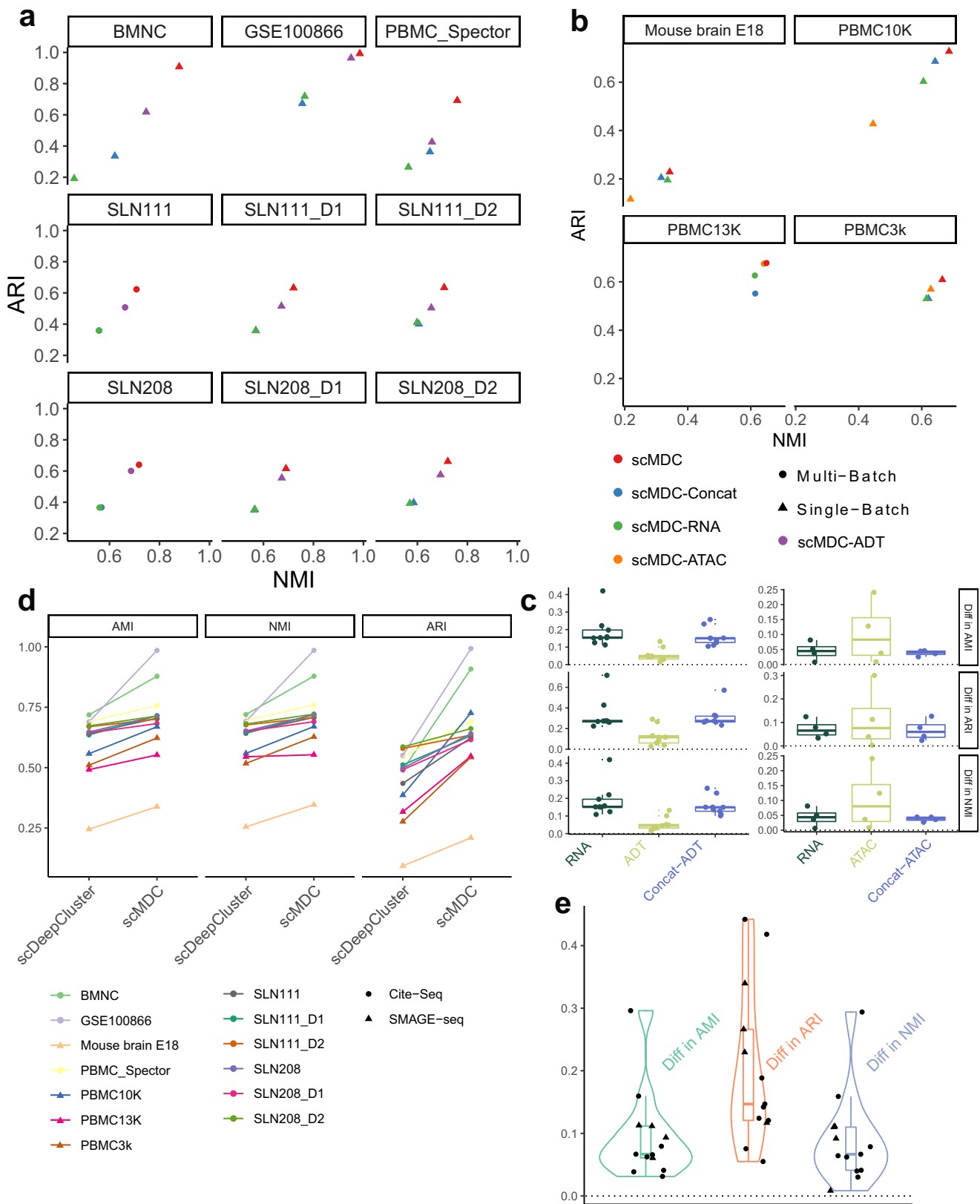

because of the low sample size of SMAGE-seq data ($n = 4$). Considering that the sub-models of scMDC are not optimized for clustering scRNA-seq data, we then compare scMDC with scDeepCluster, a state-of-art tool for clustering scRNA-seq data. It is noted that scMDC uses multi-omics data as input (either mRNA + ADT or mRNA + ATAC), while scDeepCluster only uses mRNA-seq data as input. We find that scMDC

outperforms scDeepCluster in all datasets (Fig. 7d, e), indicating that scMDC can integrate the information from multimodal data to boost clustering performance. We also build the t-SNE plots of the embeddings from scMDC and three variant models (Supplementary Fig. 3). Consolidating our expectations in the introduction, scMDC-RNA correctly separates some tiny cell types but falsely combines some large cell

**Fig. 7 | Clustering performance of scMDC and the variant models on the multimodal datasets.** scMDC, scMDC-RNA, scMDC-ADT, and scMDC-Concat are tested on the CITE-seq data (**a**, $n = 9$) and scMDC, scMDC-RNA, scMDC-ATAC, and scMDC-Concat are tested on the SMAGE-seq data (**b**, $n = 4$). In panels a and b, clustering performance is illustrated in a two-dimensional manner with ARI as the Y-axis and NMI as the X-axis. Circles stand for the results of multi-batch datasets and triangles stand for the results of single-batch datasets. The differences between the performance of scMDC and the competing methods in CITE-seq data (left, $n = 9$) and SMAGE-seq data (right, $n = 4$) are shown in boxplots (**c**). Each boxplot shows the minimum, first quartile (Q1), median, third quartile (Q3), and maximum of data.

The minimum and maximum are the smallest data point that is equal to or greater than Q1 − 1.5 * IQR and the largest data point that is equal to or less than Q3 + 1.5 * IQR, respectively. The comparisons between scMDC and scDeepCluster are shown in a dotplot (**d**, $n = 13$). The paired performance for each dataset from the two methods are connected by lines. The differences between the performance of scMDC and the scDeepCluster are shown in boxplots and violin plots (**e**, $n = 13$). The definition of boxplots is the same as that in panel (**c**). In panels (**d**) and (**e**), the results of CITE-seq data are shown by circles, and the results of SMAGE-seq data are shown by triangles. Source data is provided as a Source Data file.

types. In constrast, scMDC-ADT separates most large cell types but fails to detect some small cell types. scMDC-Concat exhibits similar performance as scMDC-RNA, which suggests a predominant role of mRNA data in the concatenated input. The t-SNE plots of SMAGE-seq data (PBMC13K) from scMDC and three variant models are shown in Supplementary Fig. 4. scMDC also outperforms the variant models in cell type partition on the latent space. In addition, we compare the single-modal scMDC (scMDC-RNA and scMDC-ADT/scMDC-ATAC) to other single-modal methods (Supplementary Figs. 5–12). We find that in most datasets, the single-modal scMDC models also have the best or close-to-best performance. Based on these single-modal methods, the multimodal scMDC further boosts the clustering performance by integrating the information from two omics of data.

## Downstream analysis

Based on the clustering results, we perform two popular downstream analyses, differential expression (DE) analysis and gene set enrichment analysis (GSEA). We employ the algorithm from ACE[36], which ranks genes based on the confidence of them to be assigned to a cluster. The DE analysis can be performed between two clusters or between one cluster and the rest of the clusters. Then, we calculate the log-fold change of each gene to get the directions of differential expression (namely upregulation or downregulation) based on the normalized mRNA counts. With gene ranks and directions, we perform GSEA to find the enriched pathways in a target cluster. Here, we show the results of the BMNC dataset (Fig. 8). We conduct DE and GSEA for the four largest clusters in the BMNC data. All comparisons are performed between the target cluster and the rest of the clusters. Figure 8a shows the DE genes for CD14 monocyte, CD4 memory T cells, CD4 naive T cells, and CD8 naive T cells. We find many proven marker genes for each cell type. For example, *LYZ, CST3, HLA-DRA, CD74,* and *CD14* have been shown to be highly expressed in the monocyte cells[37]. *CD27* and *CCR7* are the marker genes for naive cells[38]. They are in the top ranks in both CD4 naive and CD8 naive clusters. *IL7R* and *S100A4* have been demonstrated to be highly expressed in memory T cells[39]. Figure 8b shows the GSEA results of the Hallmark pathways based on the DE analysis. Hierarchical clustering is performed on both pathways and cell clusters. We find that two naive cell types are clustered together and have many common enriched pathways. The MYC targets are enriched in CD4 naive, CD4 memory, and CD8 naive clusters. Their important functions in CD4 and CD8 T cells have been demonstrated by Marchingo et al.[40]. The complement system has the highest enrichment score in CD14 monocytes. It is an essential pathway for the phagocytosis of mesenchymal stromal cells by monocytes[41]. The hypoxia pathway is enriched in CD4 memory T cells. It has been widely shown that hypoxia has a significant influence on the metabolism and differentiation of memory CD4 T cells[42–44]. IL2 signaling is also enriched in CD4 memory T cells. Its dynamic roles in CD4 T cells have been demonstrated in many previous studies[45,46]. The enrichment plots of the significant Hallmark pathways are shown in Supplementary Figs. 13–16. These downstream analyses further consolidate the correctness of the clustering results of scMDC.

## Hyperparameter tuning and time complexity

scMDC has two key hyperparameters $\varphi$(Phi) and $\gamma$(Gamma) that control the KL loss and clustering loss, respectively. Figure 9a, b shows the clustering performance of scMDC on both CITE-seq and SMAGE-seq datasets with various $\varphi$ and $\gamma$, respectively. We find that when $\varphi$ is lower than 0.01 and $\gamma$ is lower than 10, scMDC is insensitive to these parameters. When $\varphi$ goes beyond 0.01 and $\gamma$ goes beyond 10, scMDC's performance drops dramatically. It is noted that the clustering loss has a clear contribution to the performance of most datasets ($P < 0.05$ from a one-sided paired t-test between $\gamma = 0.1$ and $\gamma = 0.001$). On the other hand, the KL loss contributes slightly to the performance of some CITE-seq datasets but boosts the performance of SMAGE-seq datasets, especially in ARI. The statistical tests of the hyperparameter tuning results are listed in Supplementary Table 5.

To test the running time of scMDC, we simulate datasets with cell numbers ranging from 1000 to 100,000. Figure 9c shows the running time of scMDC with ascending cell numbers. We find a linear relationship between the cell numbers and the running time of scMDC. When the cell number is ten thousand, scMDC only needs about 7 min to finish the clustering analysis. Even when the cell number is as large as a hundred thousand, scMDC just takes about 1 h to finish the clustering analysis. All results are obtained on the Nvidia Tesla P100 with 16 Gb memory.

## Discussion

We have introduced scMDC - a multimodal deep learning method for clustering analysis of different single-cell multi-omics data. scMDC jointly models both mRNA and ADT/ATAC data by employing a multimodal autoencoder. Deep K-means clustering is conducted on the bottleneck of the autoencoder, and a KL-loss is employed to facilitate separating distinct cell groups. scMDC is an end-to-end deep model, and all components are optimized simultaneously. Current existing clustering methods for CITE-seq data either apply a shallow Bayes model, such as BREM-SC, or combine two distance-based graphs of mRNA and ADT, such as CiteFuse and Seurat, to leverage information from different data sources. These methods do not explicitly model dropout events and overdispersions in mRNA and/or ADT count data. Our real-data results demonstrate that the multimodal-based deep learning approach can characterize different sources of count data of CITE-seq and SMAGE-seq more effectively and efficiently.

The clustering results are essential for the downstream analyses, such as differential expression and gene set enrichment analysis. We employ a deep learning-based differential expression algorithm[36] to rank genes in a target cluster based on their confidence of being assigned to that cluster. Given the ranked list of genes, GSEA can be performed to profile cell types at a functional level. The advantages of this deep differential expression method over the traditional methods, such as Wilcoxon test and DEseq2[47], have been demonstrated by Lu et al.[36]. With the acceleration of GPU, scMDC is very efficient for analyzing large multi-omics datasets. Taking all results together, we conclude that scMDC is a promising method for the analysis of single-cell multi-omics data.

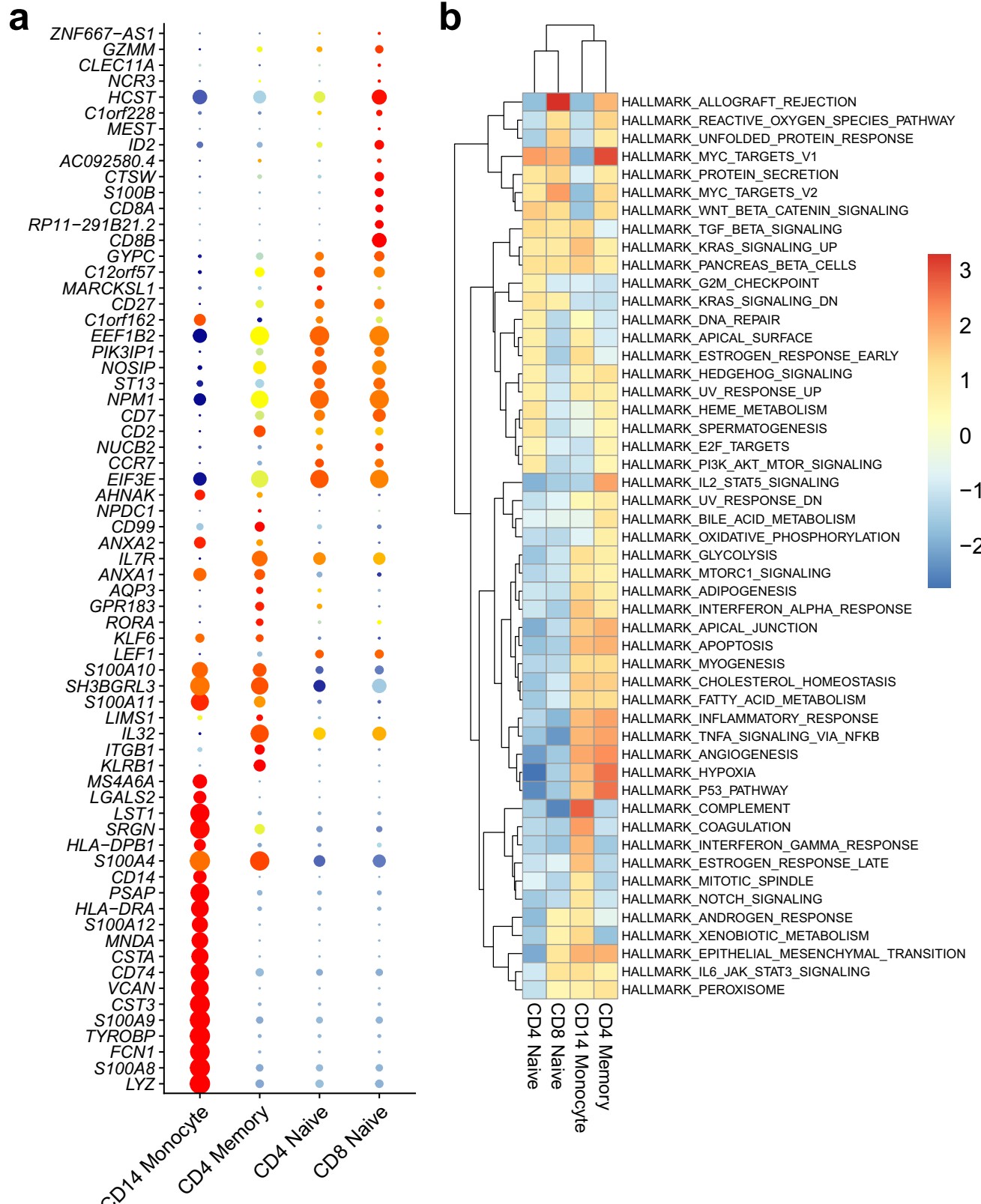

**Fig. 8 | Downstream analyses of scMDC in BMNC dataset.** Differential expression analyses (**a**) and Hallmark gene set enrichment analyses (**b**) are conducted for four large cell clusters in the BMNC dataset based on the clustering result of scMDC. In panel (**a**), the dot size shows the expression percentage of a gene in a cell type, and colors show the average expression level of a gene in a cell type with blue as low and red as high.

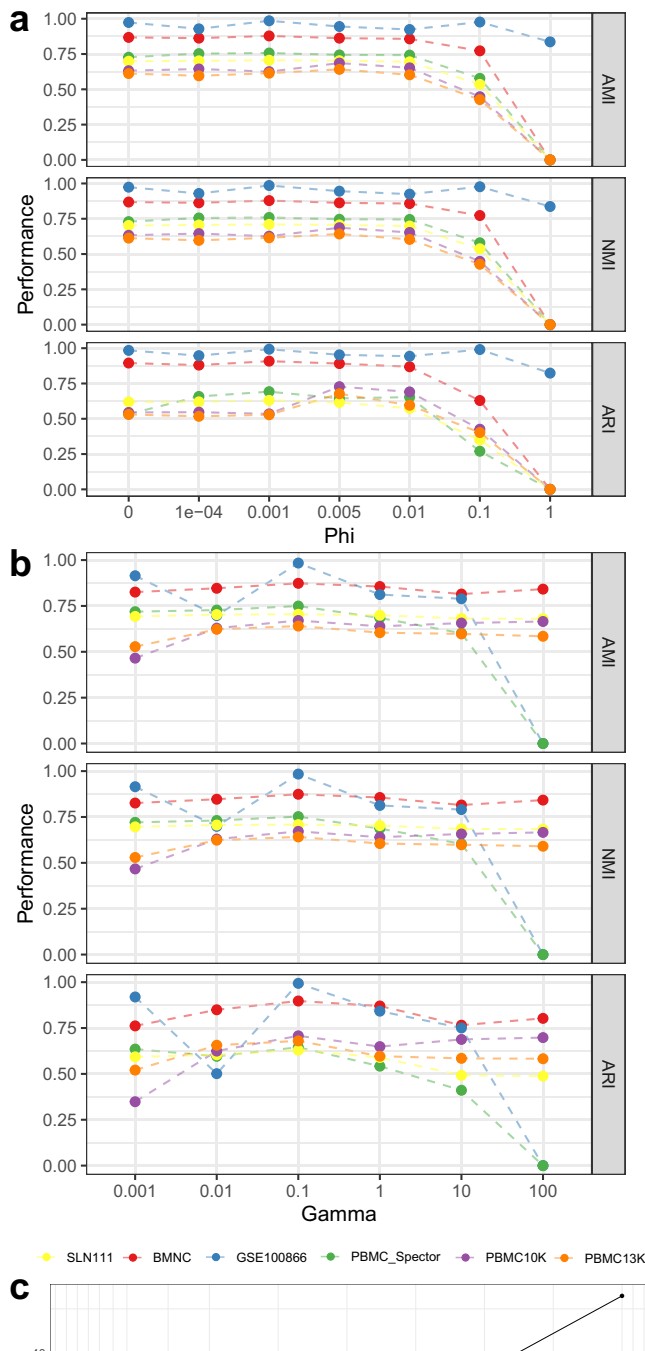

**Fig. 9 | Hyperparameter tuning and running time testing of scMDC.** This experiment is conducted on six real datasets (*n* = 6). Phi (**a**) and Gamma (**b**) are set ranging from 0 to 1 and 0.001 to 100, respectively. We test the running time of scMDC by increasing the cell numbers in the simulated datasets from 1000 to 100,000 (**c**, *n* = 7). Source data is provided as a Source Data file.

## Method

### Count data preprocessing

The raw CITE-seq data is preprocessed and normalized by the Python package SCANPY[48]. mRNA and ADT data are normalized separately but using the same method. Specifically, the genes and ADTs with no count are filtered out. The counts of a cell are normalized by a size factor $s_i$ (specifically, $s_i^p$ for ADT data and $s_i^r$ for mRNA data), which is calculated as dividing the library size of that cell by the median of the library size of all cells. In this way, all cells will have the same library size and become comparable. Finally, the counts are transformed into logarithms and scaled to have unit variance and zero mean. The treated count data of mRNA and ADT are used in our denoising multimodal autoencoder model. We use the raw count matrix to calculate the ZINB loss[30,31]. Before processing the Single-cell Multiome ATAC Gene Expression (SMAGE-seq) data, we map all the reads from scATAC-seq to the gene regions (see details below). Then we use the same methods to preprocess and normalize SMAGE-seq data as for CITE-seq data. The size factor $s_i^a$ for ATAC data is also calculated.

### Denoising hierarchical multimodal autoencoder

The autoencoder is a neural network that is able to learn nonlinear representations efficiently[49]. There are various types of autoencoder models. The denoising autoencoder receives corrupted data with artificial noises and reconstructs the original data[50]. It is widely used for noisy datasets to learn a robust latent representation. We use the denoising autoencoder for the mRNA, ADT, and ATAC data since they are very noisy. Let us denote the preprocessed counts of mRNA, ADT, and ATAC as $\mathbf{X^r}$, $\mathbf{X^p}$, and $\mathbf{X^a}$ and the corrupted mRNA, ADT and ATAC data as $\mathbf{X_c^r}$, $\mathbf{X_c^p}$, and $\mathbf{X_c^a}$, formally:

$$\mathbf{X_c^r} = \mathbf{X^r} + \sigma_r * \mathbf{n_r} \qquad (1)$$

$$\mathbf{X_c^p} = \mathbf{X^p} + \sigma_P * \mathbf{n_p} \qquad (2)$$

$$\mathbf{X_c^a} = \mathbf{X^a} + \sigma_a * \mathbf{n_a} \qquad (3)$$

here $\mathbf{n_r}$, $\mathbf{n_p}$, and $\mathbf{n_a}$ are the artificial gaussian noise (with mean = 0 and variance = 1) for mRNA, ADT and ATAC data, respectively, and $\sigma_r$, $\sigma_p$, and $\sigma_a$ controls the weights of $n_r$, $n_p$ and $n_a$. We set $\sigma_r$ and $\sigma_a$ as 2.5 and $\sigma_p$ as 1.5.

Next, ADT/ATAC and mRNA data are reduced to latent spaces by an autoencoder model. Our autoencoder model contains one encoder (*E*) for the concatenated data and two decoders (*D*) for different omics of data. Both the encoder and decoders are multi-layered fully connected neural networks. We denote encoder $\mathbf{Z} = E_\mathbf{w}(\mathbf{X_c^r} \odot \mathbf{X_c^p})$ for the concatenated mRNA and ADT data, encoder $\mathbf{Z} = E_\mathbf{w}(\mathbf{X_c^r} \odot \mathbf{X_c^a})$ for the concatenated mRNA and ATAC data, and decoder $\mathbf{X^{a'}} = D_{\mathbf{w_a'}}^a(\mathbf{Z_a})$ for ATAC data, decoder $\mathbf{X^{p'}} = D_{\mathbf{w_p'}}^p(\mathbf{Z_p})$ for ADT data, and decoder $\mathbf{X^{r'}} = D_{\mathbf{w_r'}}^r(\mathbf{Z_r})$ for mRNA data. $X^{r'}$, $X^{p'}$, and $X^{a'}$ stand for the reconstructed data of mRNA, ADT, and ATAC. **w** and **w**′ stand for the learnable weights of the encoder and the decoders, respectively. ⊙ indicates the concatenation of two matrices. The ELU activation function[51] is used for all the hidden layers in the encoder and the decoders. Batch normalization is performed on the output of all the hidden layers. The reconstruction loss functions of our autoencoder model are:

$$L_{ADT} = L\left(\mathbf{X^p}, D_{\mathbf{w_p'}}^p\left(E_\mathbf{w}(\mathbf{X_c^{con}})\right)\right) \qquad (4)$$

$$L_{ATAC} = L\left(\mathbf{X^a}, D_{\mathbf{w_a'}}^a\left(E_\mathbf{w}(\mathbf{X_c^{con}})\right)\right) \qquad (5)$$

$$L_{mRNA} = L\left(\mathbf{X^r}, D_{\mathbf{w_r'}}^r\left(E_\mathbf{w}(\mathbf{X_c^{con}})\right)\right) \qquad (6)$$

$\mathbf{X_c^{con}}$ stands for the concatenated data from either mRNA + ADT or mRNA + ATAC. For all the omics of data, we employ the zero-inflated negative binomial (ZINB) models as the reconstruction loss function[28]. It is noted that the raw count data is used in the ZINB models[28,30,31]. Let $X_{ij}^p$ be the count for cell $i$ and protein $j$ in the raw count matrix of ADT, $X_{ij}^a$ be the count for cell $i$ and gene $j$ in the raw count matrix of ATAC, and $X_{ij}^r$ be the count for cell $i$ and gene $j$ in the raw count matrix of mRNA. The NB distributions are parameterized by mean values $\mu_{ij}^p$, $\mu_{ij}^a$ and $\mu_{ij}^r$, and dispersions $\theta_{ij}^p$, $\theta_{ij}^a$ and $\theta_{ij}^r$, for ADT, ATAC and mRNA respectively. Formally:

$$NB\left(X_{ij}^p|\mu_{ij}^p,\theta_{ij}^p\right) = \frac{\Gamma\left(X_{ij}^p + \theta_{ij}^p\right)}{X_{ij}^p!\Gamma\left(\theta_{ij}^p\right)}\left(\frac{\theta_{ij}^p}{\theta_{ij}^p+\mu_{ij}^p}\right)^{\theta_{ij}^p}\left(\frac{\theta_{ij}^p}{\theta_{ij}^p+\mu_{ij}^p}\right)^{X_{ij}^p} \quad (7)$$

$$NB\left(X_{ij}^a|\mu_{ij}^a,\theta_{ij}^a\right) = \frac{\Gamma\left(X_{ij}^a + \theta_{ij}^a\right)}{X_{ij}^a!\Gamma\left(\theta_{ij}^a\right)}\left(\frac{\theta_{ij}^a}{\theta_{ij}^a+\mu_{ij}^a}\right)^{\theta_{ij}^a}\left(\frac{\theta_{ij}^a}{\theta_{ij}^a+\mu_{ij}^a}\right)^{X_{ij}^a} \quad (8)$$

$$NB\left(X_{ij}^r|\mu_{ij}^r,\theta_{ij}^r\right) = \frac{\Gamma\left(X_{ij}^r + \theta_{ij}^r\right)}{X_{ij}^r!\Gamma\left(\theta_{ij}^r\right)}\left(\frac{\theta_{ij}^r}{\theta_{ij}^r+\mu_{ij}^r}\right)^{\theta_{ij}^r}\left(\frac{\theta_{ij}^r}{\theta_{ij}^r+\mu_{ij}^r}\right)^{X_{ij}^r} \quad (9)$$

ZINB distribution is parameterized by the negative binomial of count data and an additional coefficient ($\pi_{ij}^p$, $\pi_{ij}^a$ and $\pi_{ij}^r$) for the probabilities of dropout events:

$$ZINB\left(X_{ij}^p|\mu_{ij}^p,\theta_{ij}^p,\pi_{ij}^p\right) = \pi_{ij}^p\delta_0\left(X_{ij}^p\right) + (1 - \pi_{ij}^p)NB(X_{ij}^p|\mu_{ij}^p,\theta_{ij}^p) \quad (10)$$

$$ZINB\left(X_{ij}^a|\mu_{ij}^a,\theta_{ij}^a,\pi_{ij}^a\right) = \pi_{ij}^a\delta_0\left(X_{ij}^a\right) + (1 - \pi_{ij}^a)NB(X_{ij}^a|\mu_{ij}^a,\theta_{ij}^a) \quad (11)$$

$$ZINB\left(X_{ij}^r|\mu_{ij}^r,\theta_{ij}^r,\pi_{ij}^r\right) = \pi_{ij}^r\delta_0\left(X_{ij}^r\right) + (1 - \pi_{ij}^r)NB(X_{ij}^r|\mu_{ij}^r,\theta_{ij}^r) \quad (12)$$

To estimate these parameters in the ZINB loss functions, we add three independent fully connected layers $\mathbf{M}$, $\mathbf{\theta}$, and $\mathbf{\Pi}$ to the last hidden layer of each decoder. The layers are defined as

$$\mathbf{M_{ADT}} = diag\left(s_i^p\right) \times \exp(\mathbf{w_{p(\mu)}X^{p'}}); \mathbf{\Theta_{ADT}} = \exp(\mathbf{w_{p(\theta)}X^{p'}}); \mathbf{\Pi_{ADT}} = \exp(\mathbf{w_{p(\pi)}X^{p'}}) \quad (13)$$

$$\mathbf{M_{ATAC}} = diag\left(s_i^a\right) \times \exp(\mathbf{w_{a(\mu)}X^{a'}}); \mathbf{\theta_{ATAC}} = \exp(\mathbf{w_{a(\theta)}X^{a'}}); \mathbf{\Pi_{ATAC}} = \exp(\mathbf{w_{a(\pi)}X^{a'}}) \quad (14)$$

$$\mathbf{M_{RNA}} = diag\left(s_i^r\right) \times \exp(\mathbf{w_{r(\mu)}X^{r'}}); \mathbf{\theta_{RNA}} = \exp(\mathbf{w_{r(\theta)}X^{r'}}); \mathbf{\Pi_{RNA}} = \exp(\mathbf{w_{r(\pi)}X^{r'}}) \quad (15)$$

here $\mathbf{M_{ADT}}$, $\mathbf{\theta_{ADT}}$ and $\mathbf{\Pi_{ADT}}$ are the matrices of estimated mean, dispersion, and dropout probability for the ZINB loss of ADT data, $\mathbf{M_{ATAC}}$, $\mathbf{\theta_{ATAC}}$ and $\mathbf{\Pi_{ATAC}}$ are the matrices of estimated mean, dispersion, and dropout probability for the ZINB loss of ATAC data, and $\mathbf{M_{RNA}}$, $\mathbf{\theta_{RNA}}$ and $\mathbf{\Pi_{RNA}}$ are the matrices of estimated mean, dispersion, and dropout probability for the ZINB loss of mRNA data. $\mathbf{w_{p(\mu)}}$, $\mathbf{w_{p(\theta)}}$, $\mathbf{w_{p(\pi)}}$, $\mathbf{w_{a(\mu)}}$, $\mathbf{w_{a(\theta)}}$, $\mathbf{w_{a(\pi)}}$, $\mathbf{w_{r(\mu)}}$, $\mathbf{w_{r(\theta)}}$ and $\mathbf{w_{r(\pi)}}$ are the learnable weights. The size factor $s_i^p$, $s_i^a$ and $s_i^r$ for ADT, ATAC and mRNA are calculated in the preprocessing step. The loss function of the ZINB-based autoencoder is defined as

$$L_{ADT} = \sum_{ij} -\log\left(ZINB\left(X_{ij}^p|\mu_{ij}^p,\theta_{ij}^p,\pi_{ij}^p\right)\right) \quad (16)$$

$$L_{ATAC} = \sum_{ij} -\log\left(ZINB\left(X_{ij}^a|\mu_{ij}^a,\theta_{ij}^a,\pi_{ij}^a\right)\right) \quad (17)$$

$$L_{mRNA} = \sum_{ij} -\log\left(ZINB\left(X_{ij}^r|\mu_{ij}^r,\theta_{ij}^r,\pi_{ij}^r\right)\right) \quad (18)$$

for ADT, ATAC and mRNA data, respectively.

### Conditional autoencoder

Conditional autoencoder (CAE) has been designed to integrate the data from different batches[25]. Based on the traditional autoencoder model, we add a matrix $\mathbf{B}$ on the input of the encoder and decoders. $\mathbf{B}$ is the one-hot coding from a batch vector b of cells. If there are M batches in b, the dimension of $\mathbf{B}$ would be $N \times M$. So, the encoder becomes $\mathbf{Z} = E_\mathbf{w}(\mathbf{X_c^{con}} \odot \mathbf{B})$ and the decoders become $\mathbf{X^{p'}} = D_{\mathbf{w_p}}^p(\mathbf{Z} \odot \mathbf{B})$ for ADT, $\mathbf{X^{a'}} = D_{\mathbf{w_a}}^a(\mathbf{Z} \odot \mathbf{B})$ for ATAC, and $\mathbf{X^{r'}} = D_{\mathbf{w_r}}^r(\mathbf{Z} \odot \mathbf{B})$ for mRNA data.

### Model Architecture

Our model can be used for clustering CITE-seq data and SMAGE-seq data. For CITE-seq data, the encoder is set as {256, 64, 32, 16}, the decoder for mRNA is set as {16, 64, 256} and the decoder for ADT is set as {16 20}. For SMAGE-seq data, the encoder is set as {256, 128, 64} and the decoders for both mRNA and ATAC data are set as {64, 128, 256}. So, the latent space of CITE-seq and SMAGE-seq data has 16 and 64 dimensions respectively. The overall architecture of the scMDC model is shown in Fig. 1.

### KL divergence on the latent layer

In the clustering analysis, similar points should be grouped into the same cluster. According to the method described by Chen et al.[34], we employ a KL divergence loss function to enhance the association between similar cells and prevent squeezing the centroids of clusters in the latent space. Following t-SNE[52], the t-distribution kernel function is used to describe the pairwise similarity among two cells $i$ and $i'$ in the latent space of our autoencoder:

$$q_{ii'} = \frac{(1 + |(\mathbf{Z}_i - \mathbf{Z}_{i'})||^2)^{-1}}{\sum_{l\neq i}(1 + |(\mathbf{Z}_i - \mathbf{Z}_l)||^2)^{-1}} \quad (19)$$

here $q_{ii} = 0$. The $\mathbf{P}$ is the target distribution in training, which strengthens and weakens the affinities between the cells with high and low similarities, respectively. $\mathbf{P}$ is defined as the square of $\mathbf{Q}$ then normalized:

$$p_{ii'} = \frac{q_{ii'}^2/\sum_{i\neq}q_{i'}}{\sum_{l\neq i}(q_{il}^2/\sum_{i\neq l}q_{il})} \quad (20)$$

With the two similarity distributions, we construct the KL loss function by the Kullback-Leibler (KL) divergence between $\mathbf{Q}$ and the derived target distribution $\mathbf{P}$:

$$L_{kl} = KL(\mathbf{P} \parallel \mathbf{Q}) = \sum_i \sum_j p_{ij}\log\frac{p_{ij}}{q_{ij}} \quad (21)$$

which measure the probability-distance between the two distributions. During the training process, $\mathbf{P}$ and $\mathbf{Q}$ are calculated per batch.

### Deep K-means clustering

We perform unsupervised clustering on the latent space of the autoencoder[34]. Our multimodal autoencoder learns a nonlinear

mapping for each cell $i$, which transfers two input matrices to a low-dimensional space $\mathbf{Z}$. The clustering loss function is defined as

$$L_c = \sum_{i=1}^{N} \sum_{j=1}^{K} w_{ij} \tau f(\mathbf{Z}_i, \mathbf{V}_j) \tag{22}$$

here $\mathbf{V}$ stands for the $K$ clustering centroids and $f$ calculates the Euclidean distance between a cell (in latent space) and a centroid. $\tau$ is a hyperparameter. We set $\tau$ as 1 for CITE-seq data and 0.1 for SMAGE-seq data. The Gaussian kernel function is applied in weight measuring to smooth the gradient descent optimization process:

$$\widetilde{w}_{ij} = \frac{\exp(-f(\mathbf{Z}_i, \mathbf{V}_j))}{\sum_{k=1}^{K} \exp(-f(\mathbf{Z}_i, \mathbf{V}_k))} \tag{23}$$

Then, to speed up the convergence, an inflation operation is applied on the weights:

$$w_{ij} = \frac{\widetilde{w}_{ij}^{\alpha}}{\sum_{k}^{K=1} \widetilde{w}_{ik}^{\alpha}} \tag{24}$$

here the hyperparameter $\alpha$ is set to 2.

The total loss of scMDC is defined as

$$\underset{\mathbf{w}, \mathbf{w}_p', \mathbf{w}_r', \mathbf{U}}{\mathrm{argmin}} \, L_{total}(\mathbf{X^p}, \mathbf{X^r} | \mathbf{w}, \mathbf{w}_p', \mathbf{w}_r', \mathbf{U}) = L_{mRNA}(\mathbf{X^r} | \mathbf{w}, \mathbf{w}_r') + L_{ADT}(\mathbf{X^p} | \mathbf{w}, \mathbf{w}_p')$$
$$+ \gamma * L_c(\mathbf{X^r}, \mathbf{X^p} | \mathbf{w}, \mathbf{U}) + \varphi * L_{kl}(\mathbf{X^r}, \mathbf{X^p} | \mathbf{w}) \tag{25}$$

For CITE-seq data, and

$$\underset{\mathbf{w}, \mathbf{w}_a', \mathbf{w}_r', U}{\mathrm{argmin}} \, L_{total}(\mathbf{X^a}, \mathbf{X^r} | \mathbf{w}, \mathbf{w}_a', \mathbf{w}_r', \mathbf{U}) = L_{mRNA}(\mathbf{X^r} | \mathbf{w}, \mathbf{w}_r') + L_{ATAC}(\mathbf{X^a} | \mathbf{w}, \mathbf{w}_a') + \gamma * L_c(\mathbf{X^r}, \mathbf{X^a} | \mathbf{w}, \mathbf{U})$$
$$+ \varphi * L_{kl}(\mathbf{X^r}, \mathbf{X^a} | \mathbf{w}) \tag{26}$$

For SMAGE-seq data. $\mathbf{w}$ is the weight matrix of the encoder. $\mathbf{w}_a'$, $\mathbf{w}_p'$, and $\mathbf{w}_r'$ are the weights of mRNA decoder, ADT decoder and ATAC decoder, respectively. $\mathbf{U}$ is the set of centroids initialized by K-means. Here, $\gamma$ and $\varphi$ are the hyperparameters that control the weights for the clustering loss and the KL loss, respectively. The value of $\gamma$ is set as 0.1 for all experiments. $\varphi$ is set to 0.001 for CITE-seq data and 0.005 for SMAGE-seq data.

## Marker gene detection

We employ an approach proposed by Lu et al.[36] to find marker genes in each cluster against another cluster or the rest of the clusters. Briefly, for each gene, this algorithm will find the minimal perturbation that alters the group assignment from a source group (s) to the target group(s) (t). The objective function for one-to-one comparison is:

$$\min_{\delta} ||\delta|| + \lambda \max(0, \alpha + m_s(\mathbf{x} + \delta) - m_t(\mathbf{x} + \delta)) \tag{27}$$

here the tradeoff coefficient $\lambda$ and the margin $\alpha$ are set to 100 and 1, respectively. $\mathbf{x} \in \mathbf{X}$ is the normalized data of a cell. $\delta \in \mathbb{R}^P$ is the perturbation for altering the cluster assignment of cells. L1 norm of $\delta$ is used to encourage sparsity and non-redundancy. The objective function for one-to-rest comparison is:

$$\min_{\delta} ||\delta|| + \lambda \max(0, \alpha + m_s(\mathbf{x} + \delta) - \max_{t \neq s} m_t(\mathbf{x} + \delta)) \tag{28}$$

It is equal to comparing a source cluster to a target cluster for which cell x has the highest confidence. The confidence from a cell x to

a cluster c is defined as

$$m_c(\mathbf{x}) = \log\left(\frac{\exp(-\beta \| E_\mathbf{w}(\mathbf{x}) - \mu_c \|)}{\sum_k \exp(-\beta \| E_\mathbf{w}(\mathbf{x}) - \mu_k \|)}\right) \tag{29}$$

here $\mu_c$ is the centroid of cluster c and $\beta$ is set to 1. Besides the mRNA matrix, this algorithm can also be applied to ADT and ATAC matrix.

The gene rank learned from ACE is then multiply by a direction vector of genes to get the directed gene rank. The direction vector of genes is calculated based on the log fold change between clusters by changing positive values to 1 and negative values to −1. Based on the directed gene rank, gene set enrichment analysis (GSEA) is performed by the package fgsea (v1.19.4) and msigdbr (v7.4.1) in R.

## Model implementation

The model is implemented in Python3 using PyTorch[53]. Adam with AMSGrad variant[54,55] with an initial learning rate = 0.001 is used for the pretraining stage. The Adadelta optimizer[56] with a learning rate = 1 and rho = 0.95 is used in the clustering stage. The batch size is set as 256. We pretrain the autoencoders for 400 epochs before entering the clustering stage. In the pretraining stage, we optimize the reconstruction losses in the first 200 epochs. The KL loss ($L_{kl}$) on the bottleneck layer is then added to the training in the remaining 200 epochs. After pretraining, the users need to specify the number of clusters ($K$). At the beginning of the clustering stage, we initialize $K$ centroids by implementing K-means algorithm on the pretrained latent space. During the clustering stage, all loss functions including clustering loss ($L_c$) are optimized simultaneously, and the centroids are also continuously updated by the learning process. The convergence threshold for the clustering stage is that the predicted labels are changed less than 0.1% per epoch. All experiments of scMDC in this study are conducted on Nvidia Tesla P100 (16 G) GPU.

## Competing methods

BREM-SC (v0.2.0, https://github.com/tarot0410/BREMSC)[10], CiteFuse (v1.0.0, https://github.com/SydneyBioX/CiteFuse)[20], Seurat (v4.0.4, https://github.com/satijalab/seurat)[17], IDEC (https://github.com/XifengGuo/IDEC)[33], K-means (sklearn v0.22.2, https://scikit-learn.org/stable/modules/generated/sklearn.cluster.KMeans.html), SC3 (v1.21.0, https://github.com/hemberg-lab/SC3)[18], SCVIS (v0.1.0, https://github.com/shahcompbio/scvis)[57], Tscan (v1.31.0, https://github.com/zji90/TSCAN)[15], TotalVI (scvi-tools v0.15.0, https://scvi-tools.org/), Cobolt (v1.0.0, https://github.com/epurdom/cobolt)[26], scMM (v1.0.0, https://github.com/kodaim1115/scMM)[27] and Specter (https://github.com/canzarlab/Specter)[24] are used as competing methods. For the multimodal methods, ADT/ATAC and mRNA data are used as input, and standard normalization is applied if the authors described it. For single data source methods, ADT/ATAC and mRNA matrices are preprocessed and normalized separately and then concatenated as a single input. To keep consistency, all the methods use the same highly variable genes in RNA and ATAC data and use full ADTs in the CITE-seq data. If the methods require normalized data as inputs without defining a specific way of normalization, we apply the same normalization method as that for scMDC (described above). Before doing K-means clustering, PCA is performed on the normalized mRNA data and the top 20 PCs are used for clustering. BREM-SC uses the raw count matrix as input directly. The data normalization for Citefuse follows the vignette (https://sydneybiox.github.io/CiteFuse/articles/CiteFuse.html). Specifically, mRNA counts are normalized by the function "logNormCounts" in the Scater package[58] with default settings. ADT counts are normalized and log-transformed by the function "normaliseExprs" from the CiteFuse package. Seurat uses the raw count matrices as input. Following the CITE-seq tutorial of Seurat, we use "LogNormalize" for mRNA and "centered log-ratio transformation" for

ADT data normalization. Then the function "ElbowPlot" is used to find the best PCs (principal components) for clustering. The resolution in "FindClusters" function of Seurat is adjusted for different datasets in order to estimate a satisfactory number of clusters that are close to the real $K$. For the single-omics and multi-omics clustering, the function 'FindNeighbors' and 'FindMultiModalNeighbors'[23] are used to find the neighbors of cells by the SNN (shared nearest-neighbor) and WNN (weighted nearest-neighbor) algorithms, respectively. For IDEC and Tscan, normalized data are provided as the inputs. SC3 needs both the raw data and the normalized data as input. When the cell number is higher than 5000, SC3 runs a SVM to estimate the cell types of the extra cells in a supervised manner. SCVIS is a variational autoencoder-based model aimed to reduce the dimension of scRNA-seq data. According to the author's protocol[57], the count data are firstly processed as $\log_2(\text{CPM}/10 + 1)$, where 'CPM' means the 'counts per million'. Next, we concatenate CPMs of mRNA and ADT. Then the 100 PCs are extracted from the CPM matrix by PCA and used as the input for SCVIS analysis. K-means clustering is performed on the latent output of SCVIS. For TotalVI, we keep the default setting for all the datasets according to the official pipeline (https://docs.scvi-tools.org/en/stable/tutorials/notebooks/totalVI.html). We then perform K-means clustering on the latent space from TotalVI since the number of clusters is supposed to be known. Specter[24] uses the normalized RNA and ADT expression data as the input. We used the default setting for Specter's multimodal analysis. For SMAGE-seq datasets, we compare our model to four competing methods: K-means + PCA, Seurat, scMM, and Cobolt. All the methods use the top 2000 highly variable mRNA and ATAC data from the SMAGE-seq data. If the methods need normalized data as input, we apply the same normalization method for it as that for scMDC. Before doing K-means, PCA is performed on both mRNA and ATAC data and the top 20 PCs of each are used for clustering. For Seurat, the ATAC data, which is mapped to the gene regions, is processed in the same way as for the mRNA data. Then the WNN algorithm is used for integrating multimodal data as described before. For Cobolt, we follow the official pipeline (https://github.com/epurdom/cobolt/blob/master/docs/tutorial.ipynb) to produce the data embeddings. We then perform K-means clustering on the latent space of datasets since the number of clusters is supposed to be known. We followed the tutorial provided by scMM (v1.0.0)[27] and used the default parameters. The embeddings of scMM are obtained and used for the K-means clustering.

## Evaluation metrics

Adjust Rand Index (ARI)[59], Adjusted Mutual Information (AMI)[60], and Normalized Mutual Information (NMI)[61] are used as metrics to evaluate the clustering performance.

Adjust Rand Index measures the agreements between two sets **C** and **G**. Assuming $a$ is the number of pairs of two objects in the same group in both **C** and **G**; $b$ is the number of pairs of two objects in different groups in both **C** and **G**; $c$ is the number of pairs of two objects in the same group in **C** but in different groups in **G**; and $d$ is the number of pairs of two objects in different groups in **C**, but in the same group in **G**. The ARI is defined as

$$ARI = \frac{\binom{n}{2}(a+d) - [(a+b)(a+c)+(c+d)(b+d)]}{\binom{n}{2} - [(a+b)(a+c)+(c+d)(b+d)]} \quad (30)$$

Let **C** = {$C1, C2, …, C_{tc}$} and **G** = {$G1, G2, …, G_{tg}$} be the predicted and ground truth labels on a dataset with n cells. NMI is defined as

$$NMI = \frac{I(\mathbf{C},\mathbf{G})}{\max\{H(\mathbf{C}),H(\mathbf{G})\}} \quad (31)$$

here $I(\mathbf{C},\mathbf{G})$ represents the mutual information between **C** and **G** and is defined as

$$I(\mathbf{C},\mathbf{G}) = \sum_{p=1}^{tc} \sum_{q=1}^{tg} |C_p \bigcap G_q| \log \frac{n|C_p \cap G_q|}{|C_p| \times |G_q|} \quad (32)$$

And H(C) and H(G) are the entropies:

$$H(\mathbf{C}) = -\sum_{p=1}^{tc} |C_p| \log \frac{|C_p|}{n} \quad (33)$$

$$H(\mathbf{G}) = -\sum_{p=1}^{tg} |G_p| \log \frac{|G_p|}{n} \quad (34)$$

Similarly, AMI is defined as

$$AMI(\mathbf{C},\mathbf{G}) = \frac{I(\mathbf{C},\mathbf{G}) - E\{I(\mathbf{C},\mathbf{G})\}}{\max\{H(\mathbf{C}),H(\mathbf{G})\} - E\{I(\mathbf{C},\mathbf{G})\}} \quad (35)$$

The extra component $E\{I(\mathbf{C},\mathbf{G})\}$ is the expected mutual information between two random clusters[60].

To illustrate the superiority of scMDC over the competing methods in multiple datasets, we rank the methods based on their clustering performance (AMI, NMI, and ARI) on each dataset. The lower the rank, the better the performance. Besides, a one-sided paired $t$-test is conducted to test if the clustering metrics (NMI, AMI, and ARI) of scMDC are significantly higher than that of the competing methods, which is implemented by the "t.test()" function in R. Nominal p-value <0.05 is considered to indicate a significant difference.

## Public real datasets

The real CITE-seq datasets used in this study are summarized in Supplementary Table 6. The GSE100866 dataset is downloaded from GEO (https://www.ncbi.nlm.nih.gov/geo/query/acc.cgi?acc=GSE100866). The cells in this dataset are cord blood mononuclear (CBMN) cells and annotated by Wang et al. from marker genes and ADTs[10]. Cells with 'Unknown' cell types were filtered out. The bone marrow mononuclear cells (BMNC, GSE128639) and the cell type labels are downloaded from the "bmcite" dataset in "SeuratData" package (v0.2.1). The mouse spleen lymph node datasets (SLN208 and SLN111, GSE150599) and the cell type labels are provided by TotalVI[25] on GitHub (https://github.com/YosefLab/totalVI_reproducibility). Cells are also filtered by the author. PBMC dataset is available on the 10X website (https://support.10xgenomics.com/single-cell-gene-expression/datasets). We downloaded the preprocessed data and the cell type labels from the GitHub of Specter (https://github.com/canzarlab/Specter)[24].

The real Single-cell Multiome ATAC Gene Expression (SMAGE-seq) datasets used in this study are summarized in Supplementary Table 7. All the SMAGE-seq datasets are downloaded from the 10X Genomics website (https://www.10xgenomics.com/resources/datasets). The first and second datasets are from human peripheral blood mononuclear cells (PBMCs) with about 3k and 10k cells. We denote them as PBMC3K and PBMC10K respectively. The third dataset is from the E18 mouse brain. We denote it as E18. For each dataset, mRNA counts are downloaded directly while the ATAC gene counts are generated by us. Specifically, after filtering the reads by ATAC peak region fragments, nucleosome signal, and TSS enrichment, we mapped each read to a gene region by the function 'GeneActivity' in Signac (v1.4.0)[62]. All the steps are referred to the official pipeline from Satija lab. Then, the PBMC cells are annotated by the label transferring method in Seurat V3[62] with the reference datasets "pbmc_10k_v3.rds" (https://www.dropbox.com/s/zn6khirjafoyyxl/pbmc_10k_v3.rds?dl=0) provided by Satija lab. For the E18 dataset, we

transfer the labels from another mouse brain dataset (GSE126074 P0 mouse brain cortex) and the cell type labels are provided by the author of the SNARE-seq paper[8].

## Simulation

The simulated data are generated by the R package SymSim (0.0.0.9000)[63]. The overall setting for simulation is from the Online vignettes of SymSim (https://github.com/YosefLab/SymSim/blob/master/vignettes/SymSimTutorial.Rmd). This setting was estimated from the Zeisel 2015 dataset[64]. We lower the parameter "n_de_evf" to 5 to keep about 50% differential expressed genes/ADTs in the dataset. We perform three experiments to test the clustering performance of scMDC and generate 10 datasets in each experiment. In the first experiment, we adjust the parameter "Sigma" in the function SimulateTrueCounts() to 0.6, 0.7, and 0.8 in mRNA and 0.3, 0.4, and 0.5 in ADT to simulate the high, medium, and low clustering signal among clusters (cell types). We give lower sigma values (higher signal) to ADT data than mRNA data since it has higher signal-to-noise ratios in the real datasets[10]. In the second experiment, we adjust the parameter "alpha_mean" in function True2ObservedCounts() to 0.001, 0.00075, 0.0005 in mRNA and 0.05, 0.045, 0.04 in ADT data to simulate low, medium, and high dropout rates. These settings are also consistent with the observations in the real datasets since mRNA data has higher dropout rates than ADT data as we described in the introduction. In the third experiment, we add a batch effect in the data to test the model's performance in batch effect correction. Medium signal and dropout rate are used for this data and the parameter "batch_effect_size" in function DivideBatches() is set to 1. All the simulated datasets have 8 groups, 1000 cells, 2000 genes, and 30 ADTs.

## Reporting summary

Further information on research design is available in the Nature Portfolio Reporting Summary linked to this article.

## Data availability

The GSE100866 data used in this study are available in the GEO database under accession code GSE100866. Cell type labels are downloaded from the GitHub of BREM-SC (https://github.com/tarot0410/BREMSC). The BMNC dataset and the cell type labels are downloaded from the "bmcite" dataset in "SeuratData" package (https://github.com/satijalab/seurat-data). The mouse spleen lymph node datasets (SLN208 and SLN111) and the cell type labels are provided by TotalVI[25] on GitHub (https://github.com/YosefLab/totalVI_reproducibility). These datasets are sequenced in two batches. PBMC dataset is available on 10x Genomics website (https://support.10xgenomics.com/single-cell-gene-expression/datasets) and the cell type labels are downloaded from the GitHub of Specter (https://github.com/canzarlab/Specter). All SMAGE-seq datasets (PBMC3K, PBMC10K, and mouse brain E18) are downloaded from the 10X Genomics website (https://www.10xgenomics.com/resources/datasets). Labels are transferred by Signac (v1.4.0) from the annotated datasets. Source data are provided with this paper.

## Code availability

Codes supporting this study are available on GitHub: https://github.com/xianglin226/scMDC/releases/tag/v1.0.0.

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

## Acknowledgements

This work was supported by grant R15HG012087 (Z.W.) from the National Institutes of Health (NIH), and partially supported by the National Center for Advancing Translational Sciences (NCATS), a component of NIH under award number UL1TR003017 (Z.W.). The computing resource was partially provided by Extreme Science and Engineering Discovery Environment (XSEDE) through allocation CIE160021 and CIE170034 (supported by National Science Foundation Grant No. ACI-1548562).

## Author contributions

Z.W. and H.H. conceived and supervised the project. X.L. and T.T. designed the method and conducted the experiments. X.L. and T.T. wrote the manuscript. Z.W. and H.H revised the manuscript. All authors contributed to and approved the manuscript.

## Competing interests

The authors declare no competing interests.
