## [Peer Review File · Nature Communications]

Reviewers' Comments:

Reviewer #1:

Remarks to the Author:

Lin et al. present a computational method, scMDC for clustering of CITE-Seq data. The method considers dropout events in mRNA data and uses two-layered autoencoder to learn latent features for clustering. The authors compared scMDC to several existing methods using both simulated and real CITE-Seq data. The manuscript is clearly written, and interesting results are presented. However, a number of major weaknesses are noted, including the underlying assumption about dropout in ADT data, usefulness of ADT data versus mRNA data, validity of the simulated data, statistical significance, and representativeness of the real datasets.

Major comments:

1. It is stated that ADT data does not have dropout problem. Are there any published references and/or data to support this claim?
2. It is repeatedly stated that single-cell transcriptome (mRNA) data has larger noise than protein (ADT) data. I am not aware of any study that has demonstrated this to be true. Please provide references or data to support this claim. On a related issue, it is stated that the large noise in mRNA data makes it harder to identify subtle differences of major cell types than ADT data. Given the much smaller number of features in ADT data (10-20 features in ADT data compared to thousands in mRNA data), I would expect that ADT data is less useful for identifying subtle differences than mRNA data.
3. The number of clusters in all simulation experiments is set at four. What is the rationale for it? A typical tissue type contains more than 4 cell types/subtypes. This makes the simulation unrealistic. In fact, the performance gain of scMDC is much less using real data than simulated data (comparing Figures 2 and 3).
4. Figure 4, Figure S9-10. The performance using ADT data alone is on par and in many cases better than using mRNA data. It is hard to fathom how this could be given ADT data has only 10-20 features whereas mRNA data has at least 1000 features. One possibility is that the number of clusters is only 6-7 which makes the clustering somewhat trivial.
5. The number of cell types in the three real datasets are 7, 6 and 6. There are more than 7 cell types in the human blood and bone marrow. For instance, in GSE128639 data set, there are more than 6 cell types in bone marrow. The small number of cell types makes the performance evaluation unrealistic. Along this line, all the evaluation datasets in this study are hematopoietic cells. It would be more convincing that CITE-Seq datasets on other tissues with diverse cell types are used for additional evaluation.
6. There is no statistical significance assessment in all performance metrics comparison among the methods. In many cases, it seems there is no statistically significant difference between scMDC and compared methods although the bar is slightly taller. The differences are also dependent on the performance metrics used (ARI vs. MMI vs AC).
7. Sensitivity analysis was done on hyperparameter values. What about sensitivity on neural network architecture (# nodes and layers in the autoencoders)?
8. There seems to be significant overlap between this manuscript and Tian et al. Nat. Commun. In terms of using CITE-Seq data and deep neural network approach. Please clarify.

Minor comments:

1. GSE100866 is not PBMC. It is cord blood mononuclear cells. GSE128639 is not PBMC. It is bone marrow mononuclear cells.

Reviewer #2:

Remarks to the Author:

This paper develops a multimodal deep learning model, Single Cell Multimodal Deep Clustering (scMDC), for the clustering analysis of CITE-seq data by applying three autoencoders. scMDC employs two low-level autoencoders (AEs) for characterizing mRNA and ADT data, respectively. And then scMDC obtains the integrated latent space by a high-level AEs based on the combined latent representation of the two low-level AEs. In addition, the authors provide the computational experiments (simulated and real CITE-seq datasets) to demonstrate the effectiveness of scMDC. However, there are a few of concerns in the manuscript.

1. As a bioinformatics work, comparison of clustering performance and biological analysis on a number of cases with the existing methods are important. However, in this manuscript, marker staining in tSNE representation is the only result associated with such analyses. Further work and discussion are needed, e.g., the contribution of each data type to different cell types, the related molecular characteristics and functions per data level.
2. The comparison of computational (time) efficiency is not sufficient, and it only illustrates the running efficiency of the algorithm in different sample sizes, and does not benchmark the model with other algorithms.
3. The cluster number estimation is not one of the contributions in this paper. In my opinion, KNN combined with Louvain/Leiden is a common process for scRNA-seq analysis, where the parameters of 'k' and 'resolution' are set manually and do have great impact on clustering assignment, but their combination cannot directly estimate the number of clusters. The statements in sections of Abstract and Method are not accurate/correct.
4. There are many inaccurate expressions in the Introduction section referred to background and principles of the existing methods. For instance, as mentioned above, generally KNN and Louvain could not determine the number of clusters. SC3 algorithm doesn't use spectral clustering but k-means and hierarchical clustering approaches. TotalVI uses not only a dimension reduction approach, but also deep learning model to capture the same latent space of different data types, which enables TotalVI to integrate different data types as well. In page 2 line 65~70, the authors summarized that time/memory expensive step is to construct cell-cell similarity graph or the corresponding laplacian matrix, which is inaccurate. To obtain clusters depends on similarity matrix for other methods, just as it is for your model that uses KNN to reconstruct cell-cell relationships described in the manuscript. So how could construction of similarity graph be an unbearable disadvantage for other algorithms? It should be explained clearly. Louvain/Leiden algorithm has already become one of the most popular methods for scRNA-seq clustering nowadays rather than spectral clustering. The statement is outdated. Authors should make the survey on recent advances of this area and make more accurate comments on other algorithms.
5. This article is to integrate CITE-seq, but the classic integration methods of CITE-seq (such as totalVI) are not compared.
6. In this paper, the data preprocessing and the parameter selection of other comparison algorithms are not very clear. For example, how many features are selected for comparison, how many clusters are selected in SC3 method, etc. In addition, why does SC3 algorithm produce NA cell types?
7. The purpose of this paper seems to propose a solution to the problem of single-cell data integration. However, single-data type validation does not support the conclusion that this approach is robust, and should be supplemented with other types of data (such as scRNA-seq and ATAC data).
8. Please list all tools/packages and their versions used in the analysis described in the manuscript.
9. There are flaws in the algorithm code. For example, the data in the code is unlabeled with gene and cell type. There are too many invalid prints in the documentation, such as the printing of iterative processes.

We sincerely appreciate the thorough analyses and constructive suggestions provided by the reviewers of our manuscript, which have been very helpful in guiding us to further improve our paper. In the revised manuscript, we have further strengthened our work by fully addressing all the concerns raised by reviewers. We highlighted revised parts in red in the manuscript. We hope after reading the enclosed point-by-point responses, the editor and the reviewers will concur with us that we have addressed all the raised concerns in a satisfactory manner.

The detailed point-by-point responses are as follows.

Reviewer #1

Lin et al. present a computational method, scMDC for clustering of CITE-Seq data. The method considers dropout events in mRNA data and uses two-layered autoencoder to learn latent features for clustering. The authors compared scMDC to several existing methods using both simulated and real CITE-Seq data. The manuscript is clearly written, and interesting results are presented. However, a number of major weaknesses are noted, including the underlying assumption about dropout in ADT data, usefulness of ADT data versus mRNA data, validity of the simulated data, statistical significance, and representativeness of the real datasets.

Re: We thank the reviewer for these insightful comments, which help us strengthen the manuscript.

Point-by-point responses are listed below to address each comment in detail.

Major comments:

1. It is stated that ADT data does not have dropout problem. Are there any published references and/or data to support this claim?

RE: The original paper of CITE-seq (*Stoeckius (2017) Nature methods*) states that “Cell counts based on RNA or ADT are highly correlated between both methods and this demonstrates the low dropout rate of ADT signals”. We noticed that ADT data did not have 0 expression in the real dataset (GSE100866), so we thought ADT data did not have the dropout problem. Thanks for raising this question, which made us to check more datasets carefully. We found that all ADT data in the five CITE-seq datasets used in the revised manuscript have much lower percentages of 0s ($\leq 10\%$) than mRNA data ($\geq 80\%$). Indeed, we couldn't exclude the possibility that these 0 expressions result from dropout, although low. So, we agree with the reviewer that it may be inaccurate to assume that there is no dropout event in ADT data. We have removed the inaccurate statement. Accordingly, we updated our model and used also the ZINB loss for the ADT data to account for the potential dropout events. Furthermore, we added dropout events in the simulation datasets of both mRNA and ADT to make them consistent with real datasets.

Modifications in the manuscript

- 1) We use ZINB loss to account for the dropout events in mRNA, ADT, and ATAC data. Here are the updates in the method section.

Let X_{ij}^p be the count for cell i and protein j in the raw count matrix of ADT, X_{ij}^a be the count for cell i and gene j in the raw count matrix of ATAC, and X_{ij}^r be the count for cell i and gene j in the raw count matrix of mRNA. The NB distributions are parameterized by means μ_{ij}^p , μ_{ij}^a and μ_{ij}^r , dispersions θ_{ij}^p , θ_{ij}^a and θ_{ij}^r , for ADT, ATAC and mRNA, respectively. Formally,

$$NB(X_{ij}^p | \mu_{ij}^p, \theta_{ij}^p) = \frac{\Gamma(X_{ij}^p + \theta_{ij}^p)}{X_{ij}^p! \Gamma(\theta_{ij}^p)} \left(\frac{\theta_{ij}^p}{\theta_{ij}^p + \mu_{ij}^p} \right)^{\theta_{ij}^p} \left(\frac{\mu_{ij}^p}{\theta_{ij}^p + \mu_{ij}^p} \right)^{X_{ij}^p}$$

$$NB(X_{ij}^a | \mu_{ij}^a, \theta_{ij}^a) = \frac{\Gamma(X_{ij}^a + \theta_{ij}^a)}{X_{ij}^a! \Gamma(\theta_{ij}^a)} \left(\frac{\theta_{ij}^a}{\theta_{ij}^a + \mu_{ij}^a} \right)^{\theta_{ij}^a} \left(\frac{\mu_{ij}^a}{\theta_{ij}^a + \mu_{ij}^a} \right)^{X_{ij}^a}$$

$$NB(X_{ij}^r | \mu_{ij}^r, \theta_{ij}^r) = \frac{\Gamma(X_{ij}^r + \theta_{ij}^r)}{X_{ij}^r! \Gamma(\theta_{ij}^r)} \left(\frac{\theta_{ij}^r}{\theta_{ij}^r + \mu_{ij}^r} \right)^{\theta_{ij}^r} \left(\frac{\mu_{ij}^r}{\theta_{ij}^r + \mu_{ij}^r} \right)^{X_{ij}^r}$$

The ZINB distribution is parameterized by the negative binomial of count data and an additional coefficient π_{ij}^p , π_{ij}^a and π_{ij}^r for the probabilities of dropout events:

$$ZINB(X_{ij}^p | \mu_{ij}^p, \theta_{ij}^p, \pi_{ij}^p) = \pi_{ij}^p \delta_0(X_{ij}^p) + (1 - \pi_{ij}^p) NB(X_{ij}^p | \mu_{ij}^p, \theta_{ij}^p)$$

$$ZINB(X_{ij}^a | \mu_{ij}^a, \theta_{ij}^a, \pi_{ij}^a) = \pi_{ij}^a \delta_0(X_{ij}^a) + (1 - \pi_{ij}^a) NB(X_{ij}^a | \mu_{ij}^a, \theta_{ij}^a)$$

$$ZINB(X_{ij}^r | \mu_{ij}^r, \theta_{ij}^r, \pi_{ij}^r) = \pi_{ij}^r \delta_0(X_{ij}^r) + (1 - \pi_{ij}^r) NB(X_{ij}^r | \mu_{ij}^r, \theta_{ij}^r)$$

To estimate these parameters in the ZINB loss functions, we add three independent fully-connected layers M , θ , and Π to the last hidden layer of each decoder. The layers are defined as:

$$M_{ADT} = \text{diag}(s_i^p) \times \exp(w_{p(\mu)} X^{p'}); \Theta_{ADT} = \exp(w_{p(\theta)} X^{p'}); \Pi_{ADT} = \exp(w_{p(\pi)} X^{p'})$$

$$M_{ATAC} = \text{diag}(s_i^a) \times \exp(w_{a(\mu)} X^{a'}); \Theta_{ATAC} = \exp(w_{a(\theta)} X^{a'}); \Pi_{ATAC} = \exp(w_{a(\pi)} X^{a'})$$

$$M_{RNA} = \text{diag}(s_i^r) \times \exp(w_{r(\mu)} X^{r'}); \Theta_{RNA} = \exp(w_{r(\theta)} X^{r'}); \Pi_{RNA} = \exp(w_{r(\pi)} X^{r'})$$

where M_{ADT} , Θ_{ADT} and Π_{ADT} are the matrices of estimated mean, dispersion and drop-out probability for the ZINB loss of ADT data, respectively; M_{ATAC} , Θ_{ATAC} and Π_{ATAC} are the matrices of estimated mean, dispersion and drop-out probability for the ZINB loss of ATAC data, respectively; and M_{RNA} , Θ_{RNA} and

Π_{RNA} are the matrices of estimated mean, dispersion, and dropout probability for the ZINB loss of mRNA data, respectively. $w_p(\mu)$, $w_p(\theta)$, $w_p(\pi)$, $w_a(\mu)$, $w_a(\theta)$, $w_a(\mu)$, $w_r(\mu)$, $w_r(\theta)$ and $w_r(\pi)$ are the learnable weights. The size factor s_i^p , s_i^a and s_i^r for ADT, ATAC and mRNA are calculated in the preprocessing step. The loss function of ZINB-based autoencoder is defined as:

$$L_{ADT} = \sum_{ij} -\log(\text{ZINB}(X_{ij}^p | \mu_{ij}^p, \theta_{ij}^p, \pi_{ij}^p))$$

$$L_{ATAC} = \sum_{ij} -\log(\text{ZINB}(X_{ij}^a | \mu_{ij}^a, \theta_{ij}^a, \pi_{ij}^a))$$

$$L_{mRNA} = \sum_{ij} -\log(\text{ZINB}(X_{ij}^r | \mu_{ij}^r, \theta_{ij}^r, \pi_{ij}^r))$$

for ADT, ATAC and mRNA data. The updated Figure 1 is shown below:

Fig 1. The **architecture of scMDC**. scMDC has one encoder for the concatenated data and two decoders for each modal in the multimodal data. It can be used for clustering CITE-seq data and 10X Single-Cell Multiome ATAC + Gene Expression (SMAGE-seq) data. The upper numbers on the networks are the dense layer numbers for scATAC-seq and the lower numbers are the dense layer numbers for ADT. For multi-batch datasets, scMDC will work in a conditional autoencoder manner. A one-hot batch vector will be concatenated to the input feature of the encoder and decoders. This is designed for batch effect correction. A deep K-means algorithm and a KLD loss are implemented on the bottleneck layer of the model.

2) We also added dropout events in the simulation datasets.

We simulated data using R package SymSim (0.0.0.9000)(Zhang (2019) *Nature communications*). The overall setting for simulation is from the Online vignettes of SymSim (<https://github.com/YosefLab/SymSim/blob/master/vignettes/SymSimTutorial.Rmd>). This setting was estimated from the Zeisel 2015 dataset (Zeisel (2015) *Science*). According to the real data, we lower the parameter “n_de_evf” to 5 to keep about 50% differential expressed genes/ADTs in the dataset. We performed three experiments to test the clustering performance of scMDC. In the first experiment, we adjust the parameter “Sigma” in function SimulateTrueCounts() to 0.6, 0.7, and 0.8 in mRNA and 0.3, 0.4, and 0.5 in ADT to simulate the high, medium, and low clustering signal among clusters (cell types). We give a lower sigma (higher signal) to ADT data than mRNA data since it has a stronger clustering signal in the real datasets (Wang (2020) *Nucleic acids research*). In the second experiment, we adjust the parameter “alpha_mean” in function True2ObservedCounts() to 0.001, 0.00075, 0.0005 in mRNA and 0.05, 0.045, 0.04 in ADT data to simulate low, medium, and high dropout rates. These settings are also consistent with real datasets because mRNA has higher dropout rates than ADT data. In the third experiment, we add a batch effect in the data to test the model’s performance in batch effect correction. Medium signal and dropout rate are used in this experiment and the parameter “batch_effect_size” in function DivideBatches() is set to 1. All the simulated datasets have 8 groups, 1000 cells, 2000 genes and 30 ADTs.

2. It is repeatedly stated that single-cell transcriptome (mRNA) data has larger noise than protein (ADT) data. I am not aware of any study that has demonstrated this to be true. Please provide references or data to support this claim.

RE: The larger noise in mRNA data results from its substantially higher dropout rates than the protein (ADT) data. The authors of CITE-seq have demonstrated the low dropout rate of ADT signals in their seminal work (Stoeckius (2017) *Nature methods*). To validate this claim, we also examined the five CITE-seq datasets used in this study. For comparison, we calculate the ratio of 0s for the genes profiled by both mRNA and ADT in each dataset, which is a good indicator for dropout events. As shown in the Table below, ADT has a significantly lower ratio of 0s than mRNA in all five datasets ($P < 1e-4$, Paired Wilcoxon Signed-Rank Test). These findings suggest that mRNA has much higher dropout events, which may account for its larger noises.

Dataset	MeanZeroRatioOfRNA	MeanZeroRatioOfADT	Pval
BMNC	0.904	0.003	2.02E-11
GSE100866	0.901	0.000	9.68E-05
spleen_lymph_111	0.831	0.096	1.03E-11
spleen_lymph_206	0.831	0.100	9.24E-20
PBMC	0.821	0.123	6.88E-11

The high dropout rates in mRNA data are not surprising, because genes with tiny mRNA are difficult to be captured by the technologies (Drop-seq or 10X genomics) which causes dropout events. With these large noises (resulting from dropout), many genes provide no information for clustering, or even mislead the clustering process (*Luecken (2019) Molecular systems biology*).

On a related issue, it is stated that the large noise in mRNA data makes it harder to identify subtle differences of major cell types than ADT data. Given the much smaller number of features in ADT data (10-20 features in ADT data compared to thousands in mRNA data), I would expect that ADT data is less useful for identifying subtle differences than mRNA data.

RE: For most genes, protein is the final product to fulfill their functions and messenger RNA is the immediate product. A gene's mRNA expression level may not necessarily correlate with its protein abundance because of the post-transcriptional and post-translational regulation. Thus, protein (ADT) data are ideal for characterizing cell functions and types, while mRNA data are expected to be just a (imperfect) surrogate of protein data (especially when the throughput is an issue for most protein profiling techniques). We agree that the gene coverage is the weakness of ADT data. Our point is, *when the marker genes for a cell type are included*, ADT data should be more useful in identifying subtle differences and recognizing the subpopulation of the cell type. Due to ADT's limited capacity, marker genes for known cell types are generally included first. These protein markers can sufficiently characterize their corresponding cell types, such as CD4 for CD4+ T cells, and CD8 for CD8+ T cells. With the development of CITE-seq technology, more proteins can be profiled in an experiment. For example, in the spleen lymph node datasets added in this study, 111 and 208 proteins were profiled, respectively (*Gayoso (2021) Nature Methods*). We expect that the increasing dimensionality of ADT data will allow it to identify more cell types. As stated by the authors of BREM-SC (*Wang (2020) Nucleic acids research*), "the use of cell surface proteins for cell gating is advantageous in identifying common cell types but may not successfully identify some rare cell types due to its low dimensionality."

Modifications in the manuscript

The references were added in the revised paragraph in the introduction:

It is noted that in the multimodal data, the biological information provided by different modalities is complementary (*Stoeckius (2017) Nature methods, Peterson (2017) Nature biotechnology*), and each

modality generally has its own strengths and weaknesses. Using CITE-seq as an example, its ADT modal focuses on surface proteins. ADT data have demonstrated a low dropout rate (*Stoeckius (2017) Nature methods*) and thus can reliably quantify gene activities. For the five CITE-seq datasets analyzed in this study, we observed dropout rates of up to 12% in ADT data. In contrast, there were more than 80% or even 90% zero entries in its corresponding mRNA data. For most genes, protein is the final product to fulfill their functions and messenger RNA is an immediate product. Thus, ADT data seems ideal for characterizing cell functions and types. However, due to current technique limits, ADT can profile only up to a couple of hundreds of genes. Because of this limit, investigators generally include marker genes for well-known cell types in ADT modal first. Therefore, ADT data perform better in identifying common cell types (*Stoeckius (2017) Nature methods; Wang (2020) Nucleic acids research*), such as CD4+ and CD8+ T cells, when their marker genes are profiled. However, because of its limited dimensions, ADT data may not detect rare or minor cell types well. In contrast, the full transcriptome of mRNA data can capture comprehensive cell types, but its large noise makes it difficult to distinguish cell types with subtle differences (*Stoeckius (2017) Nature methods; Luecken (2019) Molecular systems biology*), such as transitional B cells and mature B cells. The quantity of ADT and mRNA sources produced by the same gene may not be the same when considering the post-transcriptional and post-translational regulations (*Stoeckius (2017) Nature methods, Haider (2013) Current genomics*). In this case, ADT and mRNA data provide complementary information in cell type identification (*Wang (2020) Nucleic acids research*).

This statement is also supported by the embedding plots from the variant scMDC model with only mRNA or ADT input (we denoted them as scMDC-RNA and scMDC-ADT). The following figure is extracted from Figure S3 showing the t-SNE plots of the embeddings from scMDC (top), scMDC-RNA (middle), and scMDC-ADT (bottom). We can see that scMDC and scMDC-RNA can separate more small clusters from the big cell types. Compared to the scMDC-RNA, scMDC-ADT performs better on separating some big cell types. For example, we can find one or more clusters of transitional B cells on the t-SNE plots from scMDC and scMDC-ADT, but the transitional B cells are merged into the cluster of mature B cells in the t-SNE of scMDC-RNA. It is noted that scMDC can take the advantage of both scMDC-RNA and scMDC-ADT. It can both distinguish some similar cell types and detect those small and rare cell types. Besides, some cell types can only be separated from other cell types when we use both mRNA and ADT. For example, we can only find the cluster of CD122+ CD4 T cells on the t-SNE plot of scMDC. This cell type is merged with CD8 cells on the t-SNE plots of scMDC-ADT and scMDC-RNA.

3. The number of clusters in all simulation experiments is set at four. What is the rationale for it? A typical tissue type contains more than 4 cell types/subtypes. This makes the simulation unrealistic. In fact, the performance gain of scMDC is much less using real data than simulated data (comparing Figures 2 and 3).

RE: This is the setting used by BREM-SC (*Wang (2020) Nucleic acids research*), a main competing method we aim to compare with. Because the number of cell clusters was not a critical setting in our simulation experiments, we simply followed their setting. We agree that a typical tissue type contains more than 4 cell types/subtypes. To be more realistic, we have redone all the simulation experiments and increased the cluster number to 8.

Modifications in the manuscript

- 1) See the procedure of simulation in the reply to comment 1.
- 2) With the increased number of clusters, scMDC can keep superior performance over different dropout rates and clustering signals. See the results of simulation in Figure 4 attached here.

Fig 4. Clustering performance of scMDC and the competing methods on the simulation datasets. The first simulation experiment is to test the clustering performance of scMDC with low (a), medium (b), and high (c) clustering signals. The second simulation experiment is to test the clustering performance of scMDC with low (d), medium (e), and high (f) dropout rates. Since scMDC, Seurat, and TotalVI can correct the batch effect, we also test their clustering performance on a multi-batch simulation dataset (g). The averaged ranks are calculated to summarize the results in a, b, c, d, e, f, and g (h). The clustering performance is evaluated by AMI, NMI, and ARI. The error bars show the stand errors of the results.

Here are the updated simulation results in the manuscript. To test the robustness of scMDC under different scenarios, we conduct two simulation experiments with various clustering signals and dropout rates. We generate all the simulation datasets using SymSim package (v0.0.0.9) in R. Fig 4 a, b, and c show the performance of scMDC and the competing methods on the simulated CITE-seq data with low, medium, and high clustering signals, respectively. scMDC has demonstrated superior performance across all levels of clustering signals, especially in terms of AMI and NMI. TotalVI has comparable performance with scMDC in ARI, but it is outperformed by scMDC in other metrics. Besides, when the clustering signal is low, scMDC shows a greater advantage over other methods, revealing its capability to handle datasets with low signal-to-noise ratios. Fig 4 d, e, and f show the clustering results of all the methods with low, medium, and high dropout rates, respectively. We can see that scMDC yields the optimal performance under various dropout rates, followed by TotalVI. We also observe that, the higher the dropout rate, the larger improvement scMDC brings, in comparison with its competing methods. Such result is compelling because most real single-cell datasets exhibit high dropout rates. The robust

performance under high dropout events makes scMDC to be a superior clustering method. This result also consolidates our statement that scMDC is a better tool to cluster the datasets with low signal-to-noise ratios than the competing methods. For multi-batch data, we compare scMDC with TotalVI and Seurat, the only two competing methods that can correct batch effect. Medium dropout rate and clustering signal are used for simulating the multi-batch dataset. scMDC outperforms the two competing methods in all three metrics (Figure 4g). Similarly, we rank all methods in analysis of these simulated datasets. scMDC and TotalVI constantly rank No. 1 and No. 2, respectively (Figure 4h). Like the results in the real datasets, multi-omics methods have better overall performance than the single-source methods. Using one-sided t-tests, we confirm that the improvements of scMDC over competing methods are all significant (Table S3). These simulation results demonstrate that scMDC has robust clustering performance under various scenarios.

4. Figure 4, Figure S9-10. The performance using ADT data alone is on par and in many cases better than using mRNA data. It is hard to fathom how this could be given ADT data has only 10-20 features whereas mRNA data has at least 1000 features. One possibility is that the number of clusters is only 6-7 which makes the clustering somewhat trivial.

RE: As we explain in our response to your Comment 2, ADT data has its own strength in determining cell types. Due to its strength, we hold a similar view as the authors of BREM-SC (*Wang (2020) Nucleic acids research*), that “the use of cell surface proteins for cell gating is advantageous in identifying common cell types.”. Surface proteins are the gold standard for determining cell types. If ADT data could reliably quantify surface protein expression levels, it is close to the gold standard (*Stoeckius (2017) Nature methods*). In fact, BREM-SC (*Wang (2020) Nucleic acids research*) annotate major cell types using the ADT markers, which are then used as true cell labels. In the old real datasets, we just followed the cell-typing method used by BREM-SC (*Wang (2020) Nucleic acids research*). Therefore, it is not surprising that using ADT data alone could yield on par or even better performance than mRNA data. In the new study, we were able to obtain the true labels for the real datasets provided by the authors of the original papers or from other studies published in the top journals. These true labels are not simply identified by using the ADT markers but also contain more cell types. For example, in the original paper of TotalVI (*Gayoso (2021) Nature Methods*), cells were firstly clustered by Leiden on the latent space. Then the marker genes for each cluster were found by Vision (*DeTomaso et al, 2019, Nature Communications*). Cluster types were manually annotated based on the list of marker genes. Therefore, the new cell type labels contain more cell types and may be more suitable for testing the performance of models. Besides,

in the new datasets, more ADTs can be used for clustering cells. See the summary of the new datasets in the tables below.

Modifications in the manuscript

Here are the tables of the new datasets:

Table 1. Summary of the real CITE-seq datasets*

Datasets	Platform	Tissue	# of cells	# of total genes	# of ADTs	# of groups
PBMC	10X	PBMC	3,762	33,538	49	16
GSE100866	10X	CBMN	1,372	33,514	10	6
BMNC	10X	BMNC	30,672	17,009	25	27
SLN111D1	10X	SLN	9,264	13,553	112	35
SLN111D2	10X	SLN	7,564	13,553	112	35
SLN208D1	10X	SLN	8,715	13,553	209	35
SLN208D2	10X	SLN	7,105	13,553	209	35

* We selected top 1000 highly dispersed genes for experiments in all datasets

Table 2. Summary of the real Single-cell Multiome ATAC Gene Expression datasets*

Datasets	Platform	Tissue	# of cells	# of total genes	# of genes from ATAC	# of groups
PBMC3k	10X	PBMC	2,585	36,601	20,010	14
PBMC10K	10X	PBMC	11,020	36,601	20,010	12
MBE18	10X	Brain	4,780	32,285	21,807	18

* We selected top 2000 highly dispersed genes for experiments in all datasets

5. The number of cell types in the three real datasets are 7, 6 and 6. There are more than 7 cell types in the human blood and bone marrow. For instance, in GSE128639 data set, there are more than 6 cell types in bone marrow. The small number of cell types makes the performance evaluation unrealistic. Along this line, all the evaluation datasets in this study are hematopoietic cells. It would be more

convincing that CITE-Seq datasets on other tissues with diverse cell types are used for additional evaluation.

RE: Thank you for this constructive comment. Following your suggestion, we have discarded most old cell type labels and found the new cell types provided by the authors of the original paper or from other studies published in the top journals. We only kept one old dataset and labels (GSE100866), which was provided in the BREM-SC paper (*Wang (2020) Nucleic acids research*). For the PBMC and BMNC datasets, we recruited the cell type labels provided by the paper of Specter (*Ringeling (2021) Genome Research*) and Seurat-data (R package, <https://github.com/satijalab/seurat-data>). We also added two multiple-batch spleen lymph node (SLN) datasets with the cell type labels provided by the paper of TotalVI (*Gayoso (2021) Nature Methods*). The new labels of PBMC, BMNC and SLN datasets contain 16, 27 and 35 cell types, respectively. Besides CITE-Seq data, we also tested our model on three multi-omics datasets of single-cell mRNA-seq and ATAC-seq (10X Single-Cell Multiome ATAC + Gene Expression). The cell type labels of these datasets were transferred by SeuratV3 from the annotated datasets provided by the Satija lab.

Modifications in the manuscript

- 1) See the updated tables of real datasets in the reply to comment 4.
- 2) Here is the updated description of the real datasets:

The real CITE-seq datasets used in this study are summarized in **Table 1**. The GSE100866 dataset are downloaded from GEO (<https://www.ncbi.nlm.nih.gov/geo/query/acc.cgi?acc=GSE100866>). The cells in this dataset are cord blood mononuclear (CBMN) cells and annotated by Wang et al. from marker genes and ADTs (*Wang (2020) Nucleic acids research*). Cells with 'Unknown' cell types were filtered out. The bone marrow mononuclear cells (BMNC, GSE128639) and the cell type labels are all downloaded from "SeuratData" in "bmcite" dataset. The mouse spleen lymph node

datasets (SLN208 and SLN111, GSE150599) and the cell type labels are provided by TotalVI (*Gayoso (2021) Nature Methods*) in the GitHub (https://github.com/YosefLab/totalVI_reproducibility). Cells are also filtered by the author. PBMC dataset is available on the 10X website (<https://support.10xgenomics.com/single-cell-gene-expression/datasets>). We downloaded the preprocessed data and the cell type labels from the GitHub of Specter (<https://github.com/canzarlab/Specter>)(*Ringeling (2021) Genome Research*).

The real Single-cell Multiome ATAC Gene Expression (SMAGE-seq) datasets used in this study are summarized in **Table 2**. All the SMAGE-seq datasets are download from 10X Genomics website (<https://www.10xgenomics.com/resources/datasets>). The first and second datasets are from human peripheral blood mononuclear cells (PBMCs) with about 3k and 10k cells. We denote them as PBMC3K

and PBMC10K respectively. The third dataset is from E18 mouse brain. We denote it as E18. For each dataset, mRNA counts are downloaded directly while the ATAC gene counts are generated by us. Specifically, after filtering the reads by ATAC peak region fragments, nucleosome signal and TSS enrichment, we mapped each read to a gene region by the function 'GeneActivity' in Signac (Stuart (2020) *BioRxiv*). All the steps are referred to the official pipeline from Satija lab. Then, the PBMC cells are annotated by label transferring method in Seurat V3 (Stuart (2020) *BioRxiv*) with the reference datasets "pbmc_10k_v3.rds" (https://www.dropbox.com/s/zn6khirjafoyyxl/pbmc_10k_v3.rds?dl=0) provided by Satija lab. For E18 dataset, we transfer the labels from another mouse brain dataset (GSE126074 P0 mouse brain cortex) and the cell type labels are provided by the author of the SNARE-seq paper (Chen (2019) *Nature biotechnology*).

6. There is no statistical significance assessment in all performance metrics comparison among the methods. In many cases, it seem there is no statistically significant difference between scMDC and compared methods although the bar is slightly taller. The differences are also dependent on the performance metrics used (ARI vs. MMI vs AC).

RE: Thanks for this comment. Now we calculate the performance ranks of each method in different datasets; we perform a one-sided paired t-test to show the superiority of scMDC in clustering performance over the competing methods (Table S1-3). We observe that scMDC has significantly better performance than all the competing methods and in all the metrics for CITE-seq data and simulation data. For the SMAGE-seq data, scMDC still significantly outperformed all competing methods in all metrics.

Modifications in the manuscript

Updates in results section. Fig. 2c shows the averaged rank of each method for the nine datasets. We can see that scMDC constantly ranks number 1 in all datasets for all three metrics. In contrast, the second-best methods, Seurat for AMI and NMI and Specter for ARI, have an averaged rank of 3. Using one-sided t-tests, we confirm that the improvements of scMDC over competing methods are all significant (Table S1).

Fig 2c. Averaged ranks of different methods on the CITE-seq datasets

For SMAGE-seq dataset: Fig. 3c shows the averaged rank of each method for the four datasets. We can see that scMDC ranks best in all three metrics, while Cobolt is the second-best for AMI and ARI, and Seurat is the second-best for ARI. Using one-sided t-tests, we confirm that the improvements of scMDC over competing methods are all significant (Table S2).

Fig 3c. Averaged ranks of different methods on the SMAGE-seq datasets

For simulation datasets: Similarly, we rank all methods in analysis of these simulated datasets. scMDC and TotalVI constantly rank No. 1 and No. 2, respectively (Figure 4h). Like the results in the real datasets, multi-omics methods have better overall performance than the single-source methods. Using one-sided t-tests, we confirm that the improvements of scMDC over competing methods are all significant (Table S3).

Fig 4c. Averaged ranks of different methods on the simulation datasets

Here we listed the tables of P-values:

Table S1. P-values of one-sided test in CITE-seq experiments

Methods	p_AMI	p_NMI	p_ARI
BREM-SC	0.00402824	0.00355908	9.1677E-06
CiteFuse	0.0079801	0.01111861	0.00036261
IDEC	6.7698E-05	7.57E-05	5.7372E-07
Kmeans + PCA	2.1861E-05	2.1894E-05	6.0185E-05
SC3	1.4569E-05	1.366E-05	2.2145E-05
SCVIS	0.00025911	0.00030163	5.887E-06
Seurat	0.00212642	0.00220737	0.00062321
Specter	0.00015003	0.00010859	0.00161893
TotalVI	0.01579666	0.0144765	0.00069109
Tscan	2.0401E-05	2.4565E-05	1.9785E-05

Table S2. P-values of one-sided test in SMAGE-seq experiments

Methods	p_AMI	p_NMI	p_ARI
Cobolt	0.04201604	0.04327985	0.01847998
Kmeans + PCA	0.00932083	0.00834753	0.01511548
scMM	0.00944043	0.00970153	0.01339524
Seurat	0.01684468	0.01755079	0.01762545

Table S3. P-values of one-sided test in simulation experiments

Methods	p_AMI	p_NMI	p_ARI
BREMSC	0.00205187	0.00194259	6.9106E-05
CiteFuse	3.9747E-06	3.9025E-06	2.7077E-05
iDEC	5.5039E-07	5.4942E-07	9.7328E-07
PCA+Kmeans	0.00012266	0.00012191	0.00012582
SC3	7.5575E-05	7.5267E-05	9.2593E-06
SCVIS	4.3007E-05	4.2824E-05	9.334E-06
Seurat	5.6212E-06	4.9064E-06	0.00022347
Specter	1.2021E-06	1.4494E-06	1.1547E-05
TotalVI	0.0028567	0.00282413	0.0248134
Tscan	5.0904E-05	5.1844E-05	1.4705E-05

7. Sensitivity analysis was done on hyperparameter values. What about sensitivity on neural network architecture (# nodes and layers in the autoencoders)?

RE: Thank you for this comment. We have performed comprehensive studies of model testing: 1) we tested two key hyperparameters of scMDC; 2) we tested the performance of different variant models of scMDC including scMDC-RNA, scMDC-ADT, scMDC-ATAC and scMDC-Concat. The comparison between scMDC and scMDC-RNA/scMDC-ADT/scMDC-ATAC reveals the advantages of using multimodal data than single-model data. The comparison between scMDC and scMDC-concat reveals the advantages of our network structure. The number of nodes and layers were not the focus of our study, and we just followed the setting of our previous works, such as scDeepCluster (Tian et al Nature Machine Intelligence, 2019). Node and layer numbers were decreased for ADT data since it has low dimensions. We also found that slightly changing the number of nodes and layers had little impact on the clustering

performance of scMDC. This statement is supported by a previous work which was conducted to test the influences from the parameter tuning to the performance of models (*Hu (2018) BIOCOMPUTING 2019: Proceedings of the Pacific Symposium*).

Modifications in the manuscript

Here is the updated paragraph to describe the tests of model structure. As described in the introduction, different omics of data provide different and complementary information for cell clustering and cell typing. Therefore, using multi-omics data in clustering should be able to achieve better performance than using single-source data. In this experiment, we conduct two tests. In the first test, we compare the performance of scMDC with two variant models: a sub-model of scMDC with only mRNA input and reconstruction loss (named scMDC-RNA) and another sub-model of scMDC with only ADT/ATAC input and reconstruction loss (named scMDC-ADT/scMDC-ATAC). We also compare scMDC to a model with concatenated mRNA and ADT data as input but with only one reconstruction loss (named as scMDC-Concat). Fig 7 a and b show the performance of scMDC and three variant models in CITE-seq and SMAGE-seq data, respectively. We find that scMDC outperforms the variant models in all the datasets. For CITE-seq data, scMDC-ADT has the second-best performance in all datasets. This is consistent with our expectation because most ADTs are strong markers for some cell types. On the other hand, scMDC-ATAC has inferior performance in two SMAGE-seq datasets. Considering that the sub-models of scMDC are not optimized for clustering scRNA-seq data, we compare scMDC with scDeepCluster, a state-of-art tool for clustering scRNA-seq data. It is noted that scMDC uses multi-omics data as input (either mRNA + ADT or mRNA + ATAC), while scDeepCluster only uses mRNA-seq data as input. We find that scMDC outperforms scDeepCluster in all datasets (Figure 7C), indicating that scMDC can integrate the information from multimodal data to boost clustering performance. We also build the t-SNE plots of the embeddings from scMDC and three variant models (Fig S3). Consolidating our expectations in the introduction, scMDC-RNA correctly separates some tiny cell types but falsely combines some large cell types. In contrast, scMDC-ADT separates most large cell types but fails to detect some small cell types. scMDC-Concat exhibits similar performance as scMDC-RNA, which suggests a predominant role of mRNA data in the concatenate input. The t-SNE plots of SMAGE-seq data (PBMC13K) from scMDC and three variant models are shown in Fig S4. scMDC also outperforms the variant models in cell type partition on the latent space. In addition, we compare the single-modal scMDC (scMDC-RNA and scMDC-ADT/scMDC-ATAC) to other single-modal methods (Fig S5-12). We find that in most datasets, the single-modal scMDC models also have the best or close-to-best performance. Based on these single-modal methods, the multimodal scMDC further boosts the clustering performance by integrating the information from two omics of data.

Fig 7. Clustering performance of scMDC and the variant models on the multimodal datasets. scMDC, scMDC-RNA, scMDC-ADT, and scMDC-Concat are tested on the CITE-seq data (a) and scMDC, scMDC-RNA, scMDC-ATAC, and scMDC-Concat are tested on the SMAGE-seq data (b). The comparison between scMDC and scDeepCluster is shown in panel c.

8. There seems to be significant overlap between this manuscript and Tian et al. Nat. Commun. In terms of using CITE-Seq data and deep neural network approach. Please clarify.

[EDITOR: We assume it is this one: <https://www.nature.com/articles/s41467-021-22008-3>

RE: Thank you for pointing out this problem. The two methods are distinct in several aspects. First, their goals are different. scDCC is a deep learning clustering methods with constraints. It aims to cluster cells based on mRNA data, while allowing to integrate domain knowledge or information to make clustering results more interpretable. The domain information is mainly derived from marker genes. The ADT data for CITE-seq can be also used as auxiliary information, if the practitioners believe the ADT data profile the marker genes for the cell types of their interest. In contrast, scMDC employs a multimodal autoencoder for clustering multi-omics data. It uses both mRNA and ADT/ATAC data as inputs, then extract and integrate the information from different sources. In this case, scMDC can integrate different omics of data in the latent space. scMDC can also remove the batch effect for multi-batch datasets. In addition, scDCC only uses partial information in ADT data but scMDC extracts full information in ADT data. In summary, scMDC and scDCC are different in model structure, function, and application scope.

Minor comments:

GSE100866 is not PBMC. It is cord blood mononuclear cells. GSE128639 is not PBMC. It is bone marrow mononuclear cells.

Re: Thank you for the correction. We have revised these in the manuscript.

Modifications in the manuscript

Please see the updated paragraph of the real datasets in the reply to comment 5.

Reviewer #2

(Remarks to the Author):

This paper develops a multimodal deep learning model, Single Cell Multimodal Deep Clustering(scMDC), for the clustering analysis of CITE-seq data by applying three autoencoders. scMDC employs two low-level autoencoders (AEs) for characterizing mRNA and ADT data, respectively. And then scMDC obtains the integrated latent space by a high-level AEs based on the combined latent representation of the two low-level AEs. In addition, the authors provide the computational

experiments (simulated and real CITE-seq datasets) to demonstrate the effectiveness of scMDC. However, there are a few of concerns in the manuscript.

RE: We thank the reviewer for the comments, which help to strengthen the presentation of our manuscript. Our point-by-point responses are given below to address each comment in detail.

1. As a bioinformatics work, comparison of clustering performance and biological analysis on a number of cases with the existing methods are important. However, in this manuscript, marker staining in tSNE representation is the only result associated with such analyses. Further work and discussion are needed, e.g., the contribution of each data type to different cell types, the related molecular characteristics and functions per data level.

Re: Thank you for this suggestion. Following your advice, we have employed a deep learning-based differential expression (DE) approach to interpret the clustering results. Based on the DE results, we can find the marker genes in each cluster and perform a GSEA to detect the enriched pathways in specific cell types. We show the downstream analysis of the BMNC dataset in the result section. The marker genes detected by DE and the Hallmark pathways detected by GSEA are also supported by one or more existing works. Thus, the results of downstream analyses can further support that the clustering produced by scMDC makes sense.

Modifications in the manuscript

Updates in introduction. We also conduct a deep differential expression analysis and a gene set enrichment analysis after clustering. The meaningful results of these downstream analyses further support the superior clustering performance of scMDC.

Updates in method section. We employ an approach proposed by Lu et al. (Lu (2021) bioRxiv) to find marker genes in each cluster against another cluster or the rest of the clusters. Briefly, for each gene, this algorithm will find the minimal perturbation that alters the group assignment from a source group (s) to the target group(s) (t). The objective function for one-to-one comparison is:

$$\min_{\delta} \|\delta\| + \lambda \max(0, \alpha + m_s(x + \delta) - m_t(x + \delta))$$

Where the tradeoff coefficient λ and the margin α are set to 100 and 1, respectively. $x \in X$ is the normalized data of a cell. $\delta \in \mathbb{R}^P$ is the perturbation for altering the cluster assignment of cells. L1 norm of δ is used to encourage sparsity and non-redundancy. The objective function for one-to-rest comparison is:

$$\min_{\delta} \|\delta\| + \lambda \max(0, \alpha + m_s(x + \delta) - \max_{t \neq s} m_t(x + \delta))$$

It is equal to compare a source cluster to a target cluster for which the cell x has the highest confidence. The confidence from a cell x to a cluster c is defined as:

$$m_c(x) = \log\left(\frac{\exp(-\beta \|E_w(x) - \mu_c\|)}{\sum_k \exp(-\beta \|E_w(x) - \mu_k\|)}\right)$$

Where μ_c is the centroid of cluster c and β is set to 1. Besides mRNA matrix, this algorithm can also be applied to ADT and ATAC matrix.

Based on the gene rank, gene set enrichment analysis is performed by the package `fgsea` (v1.19.4) and `msigdb` (v7.4.1) in R.

Updates in result section. Based on the clustering results, we perform two popular downstream analyses, differential expression (DE) analysis and gene set enrichment analysis (GSEA). We employ the algorithm from ACE(Lu (2021) *bioRxiv*), which ranks genes based on the confidence of them to be assigned to this cluster. The DE analysis can be performed between two clusters or between one cluster and the rest of the clusters. Then, we calculate the log-fold change of each gene to get the directions of differential expression (namely upregulation or downregulation) based on the normalized mRNA counts. With gene ranks and directions, we perform GSEA to find the enriched pathways in the target clusters. Here, we show the results of the BMNC dataset (Fig 8). We conduct DE and GSEA for the four largest clusters in the BMNC data. All comparisons are performed between the target cluster and the rest of the clusters. Fig 8a shows the DE genes for CD14 monocyte, CD4 memory T cells, CD4 naive T cells, and CD8 naive T cells. We find many proven marker genes for each cell type. For example, LYZ, CST3, HLA-DRA, CD74, and CD14 have been shown highly expressed in the monocyte cells (*Schlachetzki (2018) Scientific Reports*). CD27 and CCR7 are the marker gene for naive cells (*Caccamo (2018) Frontiers in Immunology*). They are in the top ranks in both CD4 naive and CD8 naive clusters. IL7R and S100A4 have been demonstrated to be highly expressed in memory T cells (*Harding (2018) Nucleic acids research*). Fig 8b shows the GSEA results of the Hallmark pathways based on the DE analysis. Hierarchical clustering is performed on both pathways and cell clusters. We find that two naive cell types are clustered together and have many common enriched pathways. The two sets of MYC targets are enriched in CD4 naive, CD4 memory, and CD8 naive clusters. Their important functions in CD4 and CD8 T cells have been demonstrated by Marchingo et al. (*Marchingo (2020) Elife*) The complement system has the highest enrichment score in CD14 monocytes. It is an essential pathway for the phagocytosis of mesenchymal stromal cells by monocytes (*Gavin (2019) Frontiers in immunology*). The hypoxia pathway is enriched in the CD4 memory T cells. It has been widely shown that hypoxia has a significant influence on the metabolism and differentiation of memory CD4 T cells (*Cho (2019) Proceedings of the National Academy of Sciences, Dimeloe (2016) The Journal of Immunology, Hasan (2021) JCI insight*). The enrichment plots of the significant Hallmark pathways are shown in Fig S13 -S16. These downstream analyses further consolidate the correctness of the clustering results of scMDC.

Fig 8. Differential expression analysis (a) and Hallmark gene set enrichment analysis (b) for four large cell clusters in the BMNC dataset based on the clustering result of scMDC.

Updates in discussion. The clustering results are essential for the downstream analyses, such as differential expression and gene set enrichment analysis. We employ a deep learning-based differential expression algorithm(Lu (2021) *bioRxiv*) to rank genes in a target cluster based on their confidence of being assigned to the target cluster. Given the ranked list of genes, GSEA can be performed to profile cell types at a functional level. The advantages of this deep differential expression method over the traditional methods, such as Wilcoxon-test and DEseq2(Love (2014) *Genome biology*), have been demonstrated by Lu et al(Lu (2021) *bioRxiv*).

2. The comparison of computational (time) efficiency is not sufficient, and it only illustrates the running efficiency of the algorithm in different sample sizes, and does not benchmark the model with other algorithms.

Re: We do not intend to show scMDC is optimal in terms of running time. That is why we chose not to benchmark the model with other algorithms. In fact, we admit that some traditional methods, such as K-

means and Seurat, may even run faster than scMDC. Our goal is to demonstrate that scMDC has linear time complexity, and as a result, it can feasibly handle most applications and return results within an acceptable amount of time.

3. The cluster number estimation is not one of the contributions in this paper. In my opinion, KNN combined with Louvain/Leiden is a common process for scRNA-seq analysis, where the parameters of 'k' and 'resolution' are set manually and do have great impact on clustering assignment, but their combination cannot directly estimate the number of clusters. The statements in sections of Abstract and Method are not accurate/correct.

Re: Thanks for this comment. We agree that the cluster number estimation is not our contribution and we just provided it as an auxiliary tool for the user's convenience. We also agree that the combination of KNN with Louvain/Leiden cannot directly estimate the number of clusters, while the parameter 'resolution' will implicitly set the number of clusters. To avoid any confusion, we have removed the relevant statements in sections of Abstract and Method.

4. There are many inaccurate expressions in the Introduction section referred to background and principles of the existing methods. For instance, as mentioned above, generally KNN and Louvain could not determine the number of clusters. SC3 algorithm doesn't use spectral clustering but k-means and hierarchical clustering approaches. TotalVI uses not only a dimension reduction approach, but also deep learning model to capture the same latent space of different data types, which enables TotalVI to integrate different data types as well. In page 2 line 65~70, the authors summarized that time/memory expensive step is to construct cell-cell similarity graph or the corresponding laplacian matrix, which is inaccurate. To obtain clusters depends on similarity matrix for other methods, just as it is for your model that uses KNN to reconstruct cell-cell relationships described in the manuscript. So how could construction of similarity graph be an unbearable disadvantage for other algorithms? It should be explained clearly. Louvain/Leiden algorithm has already become one of the most popular methods for scRNA-seq clustering nowadays rather than spectral clustering. The statement is outdated. Authors should make the survey on recent advances of this area and make more accurate comments on other algorithms.

Re: Thank you for the suggestion. For SC3, before conducting the consensus clustering, it essentially relies on spectral clustering to obtain individual clustering results, which has all key characteristics of the

spectral clustering (Ulrike von Luxburg, *Statistics and Computing*, A tutorial on spectral clustering, 2007), especially the graph Laplacian construction followed by computing its eigenvectors, the most time-consuming step with time complexity of $O(n^3)$, then K-means clustering on the eigenvectors. The resulting consensus matrix is then clustered using hierarchical clustering at the end.

Following your advice, we have made more accurate comments on other algorithms. It is noted that the related work is focused on clustering methods for multi-omics data, so our comments on the methods for just clustering scRNA-seq data intended to be brief. As we discuss above, the cluster number estimation is not our contribution (not considered as a part of scMDC). Therefore, we do not think construction of similarity graph is the disadvantage of scMDC either. In comparison, similarity matrix/graph is required in some other methods. Anyway, we remove this controversial comment.

We agree that Louvain/Leiden algorithm has already become one of the most popular methods for scRNA-seq clustering nowadays. We added this comment with Seurat since it employs the Louvain/Leiden algorithm to do clustering.

Modifications in the manuscript

In the introduction

Seurat constructs a k-nearest neighbors (KNN) graph based on the Euclidean distance in PCA space. With the graph, it then employs the Louvain(*Blondel (2008) Journal of statistical mechanics: theory and experiment*)/Leiden algorithm to iteratively group cells together by optimizing modularity(*Butler (2018) Nature biotechnology*). The Louvain/Leiden algorithm has already become one of the most popular methods for scRNA-seq clustering. SC3 employs spectral clustering to obtain individual clustering results based on the distance matrices derived from the Euclidean, Pearson and Spearman metrics, respectively. It then computes a consensus matrix by summarizing the three individual clustering results. Finally, the consensus matrix is clustered using hierarchical clustering to produce final clustering results(*Kiselev (2017) Nature methods*).

....

Another line of research, which is relevant, focuses on learning a joint embedding of different modalities. Such joint embedding is expected to improve various downstream analyses, including clustering. TotalVI is a deep variational autoencoder which can capture the same latent space of different data types (*Gayoso (2021) Nat Methods*). With this design, totalVI can learn a joint probabilistic representation of the paired ADT and mRNA measurements from CITE-seq data that accounts for the distinct information of each modality. Similarly, for SNARE-seq or SMAGE-seq data, Cobolt (*Gong (2021) bioRxiv*) and scMM (*Minoura (2021) Cell Reports Methods*) employ a Multimodal Variational Autoencoder to jointly model the multiple modalities and learn a joint embedding of mRNA-seq and ATAC-seq data. However, these methods focusing on joint embedding are not designed and optimized for clustering, although we can,

as a naïve solution, learn joint embeddings first, which is then followed by simple clustering using, for example, k-means. Such a divided strategy is suboptimal for clustering, as shown in our experiments later.

5. This article is to integrate CITE-seq, but the classic integration methods of CITE-seq (such as totalVI) are not compared.

Re: totalVI doesn't not provide direct clustering results so we didn't include it for comparison in our original version. We agree that it is good to compare with such classic integration methods. Following your suggestion, we have added TotalVI in the updated results. In total, we added four more multi-omics methods as the competing methods including Specter and TotalVI for CITE-seq data, and scMM and Cobolt for the mRNA and ATAC multi-omics data.

Modifications in the manuscript

Here we attached the updated paragraph of competing methods for CITE-seq data analysis in the method section:

For the multimodal methods, ADT/ATAC and mRNA data are used as input, and standard normalization is applied if authors described. For single data source methods, ADT and mRNA matrices are preprocessed and normalized separately then concatenated as a single input. To keep consistency, all the methods use top 1000 highly variable genes and full ADTs of the CITE-seq data. If the methods require normalized data as inputs without defining a specific way of normalization, we apply the same normalization method as that for scMDC (described above). Before doing K-means clustering, PCA is performed on the normalized mRNA data and the top 20 PCs are used for clustering. BREM-SC uses raw count matrix as input directly. The data normalization for CiteFuse follows the vignette (<https://sydneybioX.github.io/CiteFuse/articles/CiteFuse.html>). Specifically, mRNA counts are normalized by the function "logNormCounts" in the Scater package (McCarthy (2017) *Bioinformatics*) with default settings. ADT counts are normalized and log-transformed by the function "normaliseExprs" from the CiteFuse package. Seurat uses the raw count matrices as input. Following the CITE-seq tutorial of Seurat, we use "LogNormalize" for mRNA and "centered log-ratio transformation" for ADT data normalization. Then the function "ElbowPlot" is used to find the best PCs (principal components) for clustering. The resolution in "FindClusters" function of Seurat is adjusted for different datasets in order to estimate a satisfactory number of clusters that are close to the real K . For the single-omics and multi-omics clustering, the function 'FindNeighbors' and 'FindMultiModalNeighbors' (Hao (2020) *bioRxiv*) are used to find the neighbors of cells by the SNN (shared nearest-neighbor) and WNN (weighted nearest-neighbor) algorithms, respectively. For IDEC and TScan, normalized data are provided as the inputs. SC3 needs both the raw count and the normalized count as input. The normalized data are processed in the same way with scMDC. When the cell number is higher than 5000, SC3 runs a SVM to estimate the cell types of the extra cells in a supervised manner. SCVIS is a variational autoencoder-based model aimed to

reduce the dimension of scRNA-seq data. According to the author's protocol (*Ding (2018) Nature communications*), the count data are firstly processed as $\log_2(\text{CPM}/10 + 1)$, where 'CPM' means the 'counts per million'. Next, we concatenate CPMs of mRNA and ADT. Then the 100 PCs are extracted from the CPM matrix by PCA and used as the input for SCVIS analysis. K-means clustering is performed on the latent output of SCVIS. For TotalVI, we keep the default setting for all the datasets according to the official pipeline (<https://docs.scvi-tools.org/en/stable/tutorials/notebooks/totalVI.html>). We then perform Kmeans clustering on the latent space of datasets from TotalVI since the number of clusters is supposed to be known. Specter(*Ringeling (2021) Genome Research*) uses the normalized RNA and ADT expression data as the input. We used the default setting for Specter's multimodal analysis.

6. In this paper, the data preprocessing and the parameter selection of other comparison algorithms are not very clear. For example, how many features are selected for comparison, how many clusters are selected in SC3 method, etc. In addition, why does SC3 algorithm produce NA cell types?

Re: We have added more descriptions of SC3. Briefly, we kept the same setting for all methods, such as the same normalization method and the same number of HVG genes. As explained above, SC3 relies on spectral clustering, which may run very slow when the number of cells is large. To address this issue, SC3 will first cluster up to 5000 cells. When the cell number is higher than 5000, it will sample 5000 cells for clustering, and assign NA cell types to the rest by default. SC3 provides an option to train a SVM model using the clustering labels to predict cell types for the cells not sampled. Thanks for bringing this issue to our attention. We have rerun the SC3 program and updated all the results.

Modifications in the manuscript

Here are the descriptions of SC3 to address the issues you pointed out: To keep consistency, all the methods use the same (top 1000) highly variable genes and full ADTs of the CITE-seq data. SC3 needs both the raw count and the normalized count as input. The normalized data are processed in the same way with scMDC. When the cell number is higher than 5000, SC3 runs a SVM to estimate the cell types of the extra cells in a supervised manner.

7. The purpose of this paper seems to propose a solution to the problem of single-cell data integration. However, single-data type validation does not support the conclusion that this approach is robust, and should be supplemented with other types of data (such as scRNA-seq and ATAC data).

Re: Thank you for this comment. Following your suggestion, we have added scRNA-seq and ATAC data and experiments (we called it SMAGE-seq in the manuscript). In addition, for integrating multiple datasets, our model can also remove the batch effect in the integrated latent representation for both CITE-seq and SMAGE-seq data.

Modifications in the manuscript

Updates in the introduction (some single sentences with SMAGE-seq are not listed). In addition to studying single-cell transcriptomes and surface proteins, recently, the development of single-cell approaches for the assay of the transposase accessible chromatin sequencing (scATAC-seq) provides us a chance to measure chromatin accessibility in a single cell (*Buenrostro (2015) Nature*). Specifically, these technologies are designed to identify open chromatin regions in the genome by using the hyperactive Tn5 transposase, which simultaneously tags and fragments DNA sequences in open chromatin regions (*Cusanovich (2015) Science*). The scATAC-seq enables us to explore cell type-specific biological activities and gene-to-gene regulatory networks. More recently, some multi-omics single-cell technologies have been developed to jointly profile chromatin accessibility and gene expression within a single cell (*Ma (2020) Trends in Biotechnology*), such as SNARE-seq and 10X Single-Cell Multiome ATAC + Gene Expression (we denote it as SMAGE-seq) (*Chen (2019) Nature biotechnology, Ma (2020) Cell*). Overall, these multimodal sequencing technologies provide us with a more comprehensive and complicated profile of a single cell.

Similarly, there are only a few methods that have been developed to integrate and/or cluster SNARE-seq or SMAGE-seq data. To name a few, Cobolt (*Gong (2021) bioRxiv*) and scMM (*Minoura (2021) Cell Reports Methods*) employ a Multimodal Variational Autoencoder to jointly model the multiple modalities. These models can learn a joint embedding of mRNA-seq and ATAC-seq data, which can be used for clustering analysis. The separation of the clustering process and the learning of embedding restricts these models' ability to find globally optimal clustering solutions.

Updates in results section. We then test the clustering performance of scMDC on the SMAGE-seq data. Here we compare scMDC with four competing methods: Cobolt, scMM, SeuratV4, and K-means + PCA. Cobolt and scMM are designed for multi-omics data embedding learning. SeuratV4 is developed for CITE-seq data but here we apply the WNN algorithm to the SMAGE-seq data. We test these methods on three real SMAGE-seq datasets from 10X genomics, including two PBMC datasets and one embryonic mouse brain dataset. We also conduct a multi-batch experiment by combining two PMBC datasets (denoted as PBMC13K). For scATAC-seq data, we use a cell-to-gene matrix as input for scMDC, scMM, Seurat, and Kmeans. This matrix is built by mapping ATAC reads onto the gene regions (See method for details). Cobolt uses the peak count matrix as the input. **Fig 3** shows the clustering performance of scMDC and the competing methods in single-batch datasets (a) and multi-batch datasets (b). We found that scMDC has superior performance in both single- and multi-batch datasets from all the metrics (AMI, NMI, and ARI). Cobolt is the second-best method in the tests and has a comparable performance with scMDC on the E18 dataset in AMI and NMI, but its performance is inferior to that of scMDC in other datasets. We rank all competing methods for each dataset based on their performance metrics. Fig. 3c shows the averaged rank of each method for the four datasets. We can see that scMDC ranks best in all three metrics, while Cobolt is the second-best for AMI and ARI, and Seurat is the second-best for ARI. Using one-sided t-tests, we confirm that the improvements of scMDC over competing methods are all significant (Table S2).

Fig 3. Clustering performance of scMDC and the competing methods on different SMAGE-seq datasets. All the methods are tested on three one-batch datasets (a) and one two-batch dataset (b). The averaged ranks are calculated to summarize the results in a and b (c). The error bars show the stand errors of the results. The clustering performance is evaluated by AMI, NMI, and ARI.

Updates in methods section

1) Model structure

See the descriptions of the updated model in the reply to reviewer 1 comment 1.

2) Real datasets

See the descriptions and tables of the real SMAGE-seq datasets in the reply to reviewer 1 comment 4.

3) Competing methods

For SMAGE-seq datasets, we compare our model to four competing methods: k-means + PCA, Seurat, scMM, and Cobolt. All the methods use the top 2000 highly variable mRNA and ATAC data of the SMAGE-seq data. If the methods need normalized data as input, we apply the same normalization method for it as that for scMDC. Before doing K-means, PCA is performed on both mRNA and ATAC data and the top 20 PCs of each are used for clustering. For Seurat, the ATAC data, which is mapped to the gene regions, is processed in the same way as for the mRNA data. Then WNN algorithm is used for integrating multimodal data as described before. For Cobolt, we follow the official pipeline (<https://github.com/epurdom/cobolt/blob/master/docs/tutorial.ipynb>) to produce the data embeddings. We then perform K-means clustering on the latent space of datasets since the number of clusters is supposed to be known. We followed the tutorial provided of scMM (v1.0.0)(*Minoura (2021) Cell Reports Methods*) and used the default parameters. The embeddings of scMM are obtained and used for the K-means clustering.

8. Please list all tools/packages and their versions used in the analysis described in the manuscript.

Re: We have added the version of tools in the manuscript if there are available.

Modifications in the manuscript

Here is the updated list of competing methods: BREM-SC (v0.2.0, <https://github.com/tarot0410/BREMSC>) (*Wang (2020) Nucleic acids research*), CiteFuse (v1.0.0, <https://github.com/SydneyBioX/CiteFuse>) (*Kim (2020) Bioinformatics*), Seurat (4.0.4, <https://github.com/satijalab/seurat>) (*Butler (2018) Nature biotechnology*), IDEC (<https://github.com/XifengGuo/IDEC>) (*Xie (2016) International conference on machine learning*), k-means (sklearn 0.22.2, <https://scikit-learn.org/stable/modules/generated/sklearn.cluster.KMeans.html>), SC3 (1.21.0, <https://github.com/hemberg-lab/SC3>) (*Kiselev (2017) Nature methods*), SCVIS (v0.1.0, <https://github.com/shahcompbio/scvis>) (*Ding (2018) Nature communications*), Tscan (1.31.0, <https://github.com/zji90/TSCAN>) (*Ji (2016) Nucleic acids research*), TotalVI (scvi-tools 0.15.0, <https://scvi-tools.org/>), Cobolt (v1.0.0, <https://github.com/epurdom/cobolt>)(*Gong (2021) bioRxiv*), scMM (v1.0.0, <https://github.com/kodaim1115/scMM>)(*Minoura (2021) Cell Reports Methods*) and Specter (<https://github.com/canzarlab/Specter>)(*Ringeling (2021) Genome Research*) are used as competing methods.

9. There are flaws in the algorithm code. For example, the data in the code is unlabeled with gene and cell type. There are too many invalid prints in the documentation, such as the printing of iterative processes.

Re: Thank you for the suggestions. We have updated the example data on GitHub. Besides, to avoid the redundant print, we have added a parameter in the model to control the interval of printing.

See the Modifications in the Github: <https://github.com/xianglin226/scMDC>

References

1. Stoeckius M, *et al.* Simultaneous epitope and transcriptome measurement in single cells. *Nature methods* **14**, 865-868 (2017).
2. Zhang X, Xu C, Yosef N. Simulating multiple faceted variability in single cell RNA sequencing. *Nature communications* **10**, 1-16 (2019).
3. Zeisel A, *et al.* Cell types in the mouse cortex and hippocampus revealed by single-cell RNA-seq. *Science* **347**, 1138-1142 (2015).
4. Wang X, *et al.* BREM-SC: a bayesian random effects mixture model for joint clustering single cell multi-omics data. *Nucleic acids research* **48**, 5814-5824 (2020).
5. Luecken MD, Theis FJ. Current best practices in single cell RNA-seq analysis: a tutorial. *Molecular systems biology* **15**, e8746 (2019).
6. Gayoso A, *et al.* Joint probabilistic modeling of single-cell multi-omic data with totalVI. *Nature Methods* **18**, 272-282 (2021).
7. Peterson VM, *et al.* Multiplexed quantification of proteins and transcripts in single cells. *Nature biotechnology* **35**, 936-939 (2017).
8. Haider S, Pal R. Integrated analysis of transcriptomic and proteomic data. *Current genomics* **14**, 91-110 (2013).
9. Ringeling FR, Canzar S. Linear-time cluster ensembles of large-scale single-cell RNA-seq and multimodal data. *Genome Research* **31**, 677-688 (2021).
10. Stuart T, Srivastava A, Lareau C, Satija R. Multimodal single-cell chromatin analysis with Signac. *BioRxiv*, (2020).
11. Chen S, Lake BB, Zhang K. High-throughput sequencing of the transcriptome and chromatin accessibility in the same cell. *Nature biotechnology* **37**, 1452-1457 (2019).

12. Hu Q, Greene CS. Parameter tuning is a key part of dimensionality reduction via deep variational autoencoders for single cell RNA transcriptomics. In: *BIOCOMPUTING 2019: Proceedings of the Pacific Symposium*. World Scientific (2018).
13. Lu YY, Yu T, Bonora G, Noble WS. ACE: Explaining cluster from an adversarial perspective. *bioRxiv*, (2021).
14. Schlachetzki J, *et al.* A monocyte gene expression signature in the early clinical course of Parkinson's disease. *Scientific Reports* **8**, 1-13 (2018).
15. Caccamo N, Joosten SA, Ottenhoff TH, Dieli F. Atypical human effector/memory CD4+ T cells with a naive-like phenotype. *Frontiers in Immunology*, 2832 (2018).
16. Harding SD, *et al.* The IUPHAR/BPS Guide to PHARMACOLOGY in 2018: updates and expansion to encompass the new guide to IMMUNOPHARMACOLOGY. *Nucleic acids research* **46**, D1091-D1106 (2018).
17. Marchingo JM, Sinclair LV, Howden AJ, Cantrell DA. Quantitative analysis of how Myc controls T cell proteomes and metabolic pathways during T cell activation. *Elife* **9**, e53725 (2020).
18. Gavin C, *et al.* The complement system is essential for the phagocytosis of mesenchymal stromal cells by monocytes. *Frontiers in immunology*, 2249 (2019).
19. Cho SH, *et al.* Hypoxia-inducible factors in CD4+ T cells promote metabolism, switch cytokine secretion, and T cell help in humoral immunity. *Proceedings of the National Academy of Sciences* **116**, 8975-8984 (2019).
20. Dimeloe S, *et al.* The immune-metabolic basis of effector memory CD4+ T cell function under hypoxic conditions. *The Journal of Immunology* **196**, 106-114 (2016).
21. Hasan F, Chiu Y, Shaw RM, Wang J, Yee C. Hypoxia acts as an environmental cue for the human tissue-resident memory T cell differentiation program. *JCI insight* **6**, (2021).
22. Love MI, Huber W, Anders S. Moderated estimation of fold change and dispersion for RNA-seq data with DESeq2. *Genome biology* **15**, 1-21 (2014).
23. Blondel VD, Guillaume J-L, Lambiotte R, Lefebvre E. Fast unfolding of communities in large networks. *Journal of statistical mechanics: theory and experiment* **2008**, P10008 (2008).

24. Butler A, Hoffman P, Smibert P, Papalexi E, Satija R. Integrating single-cell transcriptomic data across different conditions, technologies, and species. *Nature biotechnology* **36**, 411-420 (2018).
25. Kiselev VY, *et al.* SC3: consensus clustering of single-cell RNA-seq data. *Nature methods* **14**, 483-486 (2017).
26. Gayoso A, *et al.* Joint probabilistic modeling of single-cell multi-omic data with totalVI. *Nat Methods* **18**, 272-282 (2021).
27. Gong B, Zhou Y, Purdom E. Cobolt: Joint analysis of multimodal single-cell sequencing data. *bioRxiv*, (2021).
28. Minoura K, Abe K, Nam H, Nishikawa H, Shimamura T. A mixture-of-experts deep generative model for integrated analysis of single-cell multiomics data. *Cell Reports Methods* **1**, 100071 (2021).
29. McCarthy DJ, Campbell KR, Lun AT, Wills QF. Scater: pre-processing, quality control, normalization and visualization of single-cell RNA-seq data in R. *Bioinformatics* **33**, 1179-1186 (2017).
30. Hao Y, *et al.* Integrated analysis of multimodal single-cell data. *bioRxiv*, (2020).
31. Ding J, Condon A, Shah SP. Interpretable dimensionality reduction of single cell transcriptome data with deep generative models. *Nature communications* **9**, 1-13 (2018).
32. Buenrostro JD, *et al.* Single-cell chromatin accessibility reveals principles of regulatory variation. *Nature* **523**, 486-490 (2015).
33. Cusanovich DA, *et al.* Multiplex single-cell profiling of chromatin accessibility by combinatorial cellular indexing. *Science* **348**, 910-914 (2015).
34. Ma A, McDermaid A, Xu J, Chang Y, Ma Q. Integrative methods and practical challenges for single-cell multi-omics. *Trends in Biotechnology*, (2020).
35. Ma S, *et al.* Chromatin potential identified by shared single-cell profiling of RNA and chromatin. *Cell* **183**, 1103-1116. e1120 (2020).
36. Kim HJ, Lin Y, Geddes TA, Yang JYH, Yang P. CiteFuse enables multi-modal analysis of CITE-seq data. *Bioinformatics* **36**, 4137-4143 (2020).

37. Xie J, Girshick R, Farhadi A. Unsupervised deep embedding for clustering analysis. In: *International conference on machine learning* (2016).
38. Ji Z, Ji H. TSCAN: Pseudo-time reconstruction and evaluation in single-cell RNA-seq analysis. *Nucleic acids research* **44**, e117-e117 (2016).

Reviewers' Comments:

Reviewer #1:

Remarks to the Author:

Here are my comments to the authors' responses. Although some issues are addressed, several other major issues still remain.

1. It is stated that ADT data does not have dropout problem. Are there any published references and/or data to support this claim?

The authors have addressed my comment.

2. It is repeatedly stated that single-cell transcriptome (mRNA) data has larger noise than protein (ADT) data. I am not aware of any study that has demonstrated this to be true. Please provide references or data to support this claim.

The authors equate dropout rate to noise, which I disagree. Dropout certainly is a source of noise but there are other types of noises, such as false positives. Therefore, if the authors just mean dropout then they should use the word "noise" to be more precise.

On a related issue, it is stated that the large noise in mRNA data makes it harder to identify subtle differences of major cell types than ADT data. Given the much smaller number of features in ADT data (10-20 features in ADT data compared to thousands in mRNA data), I would expect that ADT data is less useful for identifying subtle differences than mRNA data.

Authors stated "when the marker genes for a cell type are included, ADT data should be more useful in identifying subtle differences and recognizing the subpopulation of the cell type".

I still disagree. For instance for CD4+ vs CD8+ both protein and mRNA information can distinguish the two cell types well. The problem is comparing to mRNA, the number of proteins is still much smaller even though the new CITE-Seq datasets have antibodies targeting over 100 proteins.

Additionally, the examples given in figure S3 do not support their claim. For instance, I do not see more than 1 clusters representing transitional B cells in all three approaches (scMDC, scMDC-RNA, scMDC-ADT). There is also no CD122+ CD4 cluster shown in figure S3. By the way, what is the function of CD122+ CD4 T cells?

3. The number of clusters in all simulation experiments is set at four. What is the rationale for it? A typical tissue type contains more than 4 cell types/subtypes. This makes the simulation unrealistic. In fact, the performance gain of scMDC is much less using real data than simulated data (comparing Figures 2 and 3).

Are the raw performance values (not the ranks) in the new figure 4a-g statistically significant between scMDC and other methods? If they are not then the averaged ranks in figure 4h is an artefact of introducing the ranking. This comment applies to all figures. ie, when the authors show the raw performance metrics, they need to show if the differences are statistically significant or not.

4. Figure 4, Figure S9-10. The performance using ADT data alone is on par and in many cases better than using mRNA data. It is hard to fathom how this could be given ADT data has only 10-20 features whereas mRNA data has at least 1000 features. One possibility is that the number of clusters is only 6-7 which makes the clustering somewhat trivial.

Please see my comment to the response to my previous comment #2. This issue remains.

5. The number of cell types in the three real datasets are 7, 6 and 6. There are more than 7 cell types in the human blood and bone marrow. For instance, in GSE128639 data set, there are more

than 6 cell types in bone marrow. The small number of cell types makes the performance evaluation unrealistic. Along this line, all the evaluation datasets in this study are hematopoietic cells. It would be more convincing that CITE-Seq datasets on other tissues with diverse cell types are used for additional evaluation.

I appreciate the authors' effort to improve the cell type annotations to make the analysis more realistic. However, as stated in my comment #3, are the performance metric differences statistically significant using these new datasets and annotations?

6. There is no statistical significance assessment in all performance metrics comparison among the methods. In many cases, it seem there is no statistically significant difference between scMDC and compared methods although the bar is slightly taller. The differences are also dependent on the performance metrics used (ARI vs. MMI vs AC).

Looks like the t-test is on the ranks. This is not correct. The statistical tests need to be done on the raw performance metrics.

7. Sensitivity analysis was done on hyperparameter values. What about sensitivity on neural network architecture (# nodes and layers in the autoencoders)?

Again, the statistical tests need to be done on the raw performance metrics.

8. There seems to be significant overlap between this manuscript and Tian et al. Nat. Commun. In terms of using CITE-Seq data and deep neural network approach. Please clarify.

I appreciate the clarification. Methodology wise, the two manuscript still bear significant similarity.

Reviewer #2:

Remarks to the Author:

No further comment.

REVIEWER COMMENTS

Reviewer #1 (Remarks to the Author):

Here are my comments to the authors' responses. Although some issues are addressed, several other major issues still remain.

Re: We really appreciate the time and effort the reviewer has dedicated to providing insightful feedback. Please see our point-to-point replies below.

1. It is stated that ADT data does not have dropout problem. Are there any published references and/or data to support this claim?

The authors have addressed my comment.

2. It is repeatedly stated that single-cell transcriptome (mRNA) data has larger noise than protein (ADT) data. I am not aware of any study that has demonstrated this to be true. Please provide references or data to support this claim.

The authors equate dropout rate to noise, which I disagree. Dropout certainly is a source of noise but there are other types of noises, such as false positives. Therefore, if the authors just mean dropout then they should use the word "noise" to be more precise.

RE: We agree that there are other types of noises, such as false positives. The dropout events are so pervasive that, in our opinion, this source of noise is dominant. Nevertheless, it is hard to quantify and compare the level of different noise sources and its impact to downstream analysis. It is not the focus of this study either. Therefore, following your suggestion, we use 'dropout' directly, to be more precise.

On a related issue, it is stated that the large noise in mRNA data makes it harder to identify subtle differences of major cell types than ADT data. Given the much smaller number of features in ADT data (10-20 features in ADT data compared to thousands in mRNA data), I would expect that ADT data is less useful for identifying subtle differences than mRNA data.

Authors stated "when the marker genes for a cell type are included, ADT data should be more useful in identifying subtle differences and recognizing the subpopulation of the cell type".

I still disagree. For instance for CD4+ vs CD8+ both protein and mRNA information can distinguish the two cell types well. The problem is comparing to mRNA, the number of proteins is still much smaller even though the new CITE-Seq datasets have antibodies targeting over 100 proteins.

RE: We agree that both protein and mRNA information of the marker genes can help to distinguish the two cell types. We think there are two important factors that can impact their usefulness and would implicate the actual clustering performance. First, the number of irrelevant features that would impair the learning. For the supervised learning, we refer the reviewer to (Fan and Li, Sure Independence Screening for Ultra-High Dimensional Feature Space, *Journal of the Royal Statistical Society: Series B*, 2018). For unsupervised learning, e.g., clustering here, we hypothesize that the high dimensionality with irrelevant features would hurt learning performance similarly. While the transcriptome profiled by RNA-seq (mRNA) may cover most genes for a certain cell type, it will also include genes irrelevant for distinguishing the cell types. We have an ongoing research project in which we aim to design a novel method to perform feature selection and clustering simultaneously. Through simulation studies, we found that when the number of irrelevant genes was large enough, the clustering performance started to decrease significantly, for example, from ARI=0.67 using just all marker genes vs ARI=0.35 when several hundreds of irrelevant genes were added (NMI/Accuracy had similar patterns).

Second, dropout noise (false negatives) in mRNA may be more damaging than other sources of noises in protein data, e.g., false positives resulting from background. We had this speculation because in microarray study, the most popular normalization algorithm RMA is able to perform optimal normalization using only perfect match probes (Irizarry et al, Exploration, normalization, and summaries of high-density oligonucleotide array probe level data, *Biostatistics*, 2003). In contrast, when dropout rates are very high, imputation may not rescue false negatives (well).

Nevertheless, we agree to disagree, because it is indeed hard to prove/disapprove which is more useful due to the noise implication. Again, this is not a focus of this study either. Therefore, we now state these two kinds of data can provide complementary information instead claiming which is more useful.

Additionally, the examples given in figure S3 do not support their claim. For instance, I do not see more than 1 clusters representing transitional B cells in all three approaches (scMDC, scMDC-RNA, scMDC-ADT). There is also no CD122+ CD4 cluster shown in figure S3. By the way, what is the function of CD12++ CD4 T cells?

RE: We really apologize for the typo and unclear statements. The cell type we indicated is CD122+ CD8+, not CD122+ CD4+. We have attached an updated figure below, which shows the positions of the mentioned cell types. Specifically, transitional B cells are highlighted by the red circles; CD122+ CD8 T cells are highlighted by the blue circles. We find that 1) scMDC-RNA fails to separate transitional and mature B cells; 2) scMDC-ADT fails to separate CD8+ and CD122+ CD8+ T cells; 3) by integrating the two data sources, scMDC differentiates both the transitional and mature B cells, and the CD8+ and CD122+ CD8+ T cells, respectively. By the way, CD122+ CD8+ T cells regulate T cell homeostasis, and act as regulatory T-cells (Treg) by suppressing both autoimmune and alloimmune responses (Liu (2015) *Frontiers in immunology*).

3. The number of clusters in all simulation experiments is set at four. What is the rationale for it? A typical tissue type contains more than 4 cell types/subtypes. This makes the simulation unrealistic. In fact, the performance gain of scMDC is much less using real data than simulated data (comparing Figures 2 and 3).

Are the raw performance values (not the ranks) in the new figure 4a-g statistically significant between scMDC and other methods? If they are not then the averaged ranks in figure 4h is an artefact of introducing the ranking. This comment applies to all figures. ie, when the authors show the raw performance metrics, they need to show if the differences are statistically significant or not.

RE: Yes. It is the raw performance values that were compared, and they were statistically significant between scMDC and other methods, as shown in the supplementary tables. There were some typos in the method description which might lead to confusion. We apologize for the misleading statements and have clarified the relevant analyses in all places. In addition, we have added the boxplots of the paired differences of the raw performance values between scMDC and the competing methods (see Figs. 2, 3, 4 and 7).

4. Figure 4, Figure S9-10. The performance using ADT data alone is on par and in many cases better than using mRNA data. It is hard to fathom how this could be given ADT data has only 10-20 features whereas mRNA data has at least 1000 features. One possibility is that the number of clusters is only 6-7 which makes the clustering somewhat trivial.

Please see my comment to the response to my previous comment #2. This issue remains.

RE: Please see our response to your previous comment #2. This is possible in our opinion, for example, when the 1000+ features of mRNA data are mostly genes irrelevant to the cell types and/or the marker genes suffer severely from dropout noises, while ADT data may cover the marker genes well for the cell types in the dataset despite its small number of features. Take the dataset GSE100866, which has only 10 ADT features, for example. This dataset has 6 cell types which are annotated based on the 8 genes out of the 10 ADT features. Even these 8 genes are covered by mRNA data, separating their signals from several hundreds of other genes may be hard and the high dimensionality with sparse signal and pervasive dropouts may be a curse.

5. The number of cell types in the three real datasets are 7, 6 and 6. There are more than 7 cell types in the human blood and bone marrow. For instance, in GSE128639 data set, there are more than 6 cell types in bone marrow. The small number of cell types makes the performance evaluation unrealistic. Along this line, all the evaluation datasets in this study are hematopoietic cells. It would be more convincing that CITE-Seq datasets on other tissues with diverse cell types are used for additional evaluation.

I appreciate the authors' effort to improve the cell type annotations to make the analysis more realistic. However, as stated in my comment #3, are the performance metric differences statistically significant using these new datasets and annotations?

RE: Yes, the performance metric differences are statistically significant using these new datasets and annotations. See our response to comment #3. The comparison was based on the performance metrics (NMI, AMI, and ARI) (See Sup. Tables).

6. There is no statistical significance assessment in all performance metrics comparison among the methods. In many cases, it seem there is no statistically significant difference between scMDC and compared methods although the bar is slightly taller. The differences are also dependent on the performance metrics used (ARI vs. MMI vs AC).

Looks like the t-test is on the ranks. This is not correct. The statistical tests need to be done on the raw performance metrics.

RE: The statistical tests were done on the raw performance metrics, and we apologize for the misleading statements, which have been fixed in the revised manuscript.

7. Sensitivity analysis was done on hyperparameter values. What about sensitivity on neural network architecture (# nodes and layers in the autoencoders)?

Again, the statistical tests need to be done on the raw performance metrics.

RE: We thank the reviewer for this constructive advice. We have added the statistical test of this section and attached the results in the supplementary materials. Specifically, for a given parameter, we use the model with the lowest value of this parameter as the benchmark. Then, we run the models with incremental values of this parameter and compare their clustering performance (AMI, NMI, and ARI) to the benchmark model by using a one-sided paired t-test. We find that the clustering loss (Gamma) can significantly boost the performance of the model when it is set to 0.1 (P-value=0.02), while the improvements from the KL loss (Phi) are marginally significant (P-value=0.06).

8. There seems to be significant overlap between this manuscript and Tian et al. Nat. Commun. In terms of using CITE-Seq data and deep neural network approach. Please clarify.

I appreciate the clarification. Methodology wise, the two manuscript still bear significant similarity.

RE: We thank the reviewer for this comment. While the methodologies in the two manuscripts bear similarity, we think the differences between them are critical and significant, as highlighted below:

- 1) Different model architecture. scMDC employs a multimodal autoencoder with one encoder and two decoders, while scDCC employs an autoencoder model with one encoder and one decoder.
- 2) Different clustering algorithm. scMDC employs a deep k-means clustering algorithm, while scDCC employs a deep embedding clustering algorithm.
- 3) Additional components. scMDC employs a KL loss to help the learning of embedding from multi-omics data.
- 4) More functions: scMDC provides quite a few additional new functions for 1) multi-batch data analysis; 2) various multi-omics data analysis (CITE-seq and SMAGE-seq); and 3) downstream DE analysis (by transplanting ACE (Lu (2021) International Conference on Machine Learning) to scMDC), which are not available in scDCC.

Reviewers' Comments:

Reviewer #1:

Remarks to the Author:

The authors have addressed all my comments.

The figure legends are missing.